# Learning Cut Generating Functions for Integer Programming

**Hongyu Cheng**
Dept. of Applied Mathematics & Statistics
Johns Hopkins University
Baltimore, MD 21218
`hongyucheng@jhu.edu`

**Amitabh Basu**
Dept. of Applied Mathematics & Statistics
Johns Hopkins University
Baltimore, MD 21218
`basu.amitabh@jhu.edu`

## Abstract

The branch-and-cut algorithm is the method of choice to solve large scale integer programming problems in practice. A key ingredient of branch-and-cut is the use of *cutting planes* which are derived constraints that reduce the search space for an optimal solution. Selecting effective cutting planes to produce small branch-and-cut trees is a critical challenge in the branch-and-cut algorithm. Recent advances have employed a data-driven approach to select good cutting planes from a parameterized family, aimed at reducing the branch-and-bound tree size (in expectation) for a given distribution of integer programming instances. We extend this idea to the selection of the best cut generating function (CGF), which is a tool in the integer programming literature for generating a wide variety of cutting planes that generalize the well-known Gomory Mixed-Integer (GMI) cutting planes. We provide rigorous sample complexity bounds for the selection of an effective CGF from certain parameterized families that provably performs well for any specified distribution on the problem instances. Our empirical results show that the selected CGF can outperform the GMI cuts for certain distributions. Additionally, we explore the sample complexity of using neural networks for instance-dependent CGF selection.

## 1 Introduction

Integer linear programming is an optimization framework that has diverse applications in logistics [Arntzen et al., 1995, Sinha et al., 1995, Hane et al., 1995], finance [Bertsimas et al., 1999], engineering [Grossmann and Kravanja, 1995], national defense [Gryffenberg et al., 1997, Jiang et al., 2014], healthcare [Ajayi et al., 2024, Valeva et al., 2023], statistics [Bertsimas et al., 2016, Dash et al., 2018, Wei et al., 2019] and machine learning [Bertsimas and Dunn, 2017, Chen et al., 2021, Malioutov et al., 2023], to give a small sample of applications and related literature. The method of choice for solving integer programming problems is the well-known *branch-and-cut* paradigm [Schrijver, 1986, Nemhauser and Wolsey, 1988, Conforti et al., 2014]. This procedure has two critical ingredients: *branching*, which is a way to subdivide the problem into smaller subproblems, and the use of *cutting planes* which is a way to reduce the feasible region of a (sub)problem by deriving additional linear constraints that are implicitly implied by the integrality of the decision variables. To get to a concrete algorithm from this high level framework, one needs to make certain choices (such as how to branch, which cutting plane to use etc.) Thus, at a high level, the branch-and-cut method is really a collection of algorithms equipped with a common set of "tunable parameters". Upon a specific choice of values for these parameters, one actually obtains a well-defined algorithm that one can deploy on instances of the problem.

38th Conference on Neural Information Processing Systems (NeurIPS 2024).

There is a substantial amount of literature on the mathematical foundations of cutting planes and branching, and many of their theoretical properties are well understood. Nevertheless, current theory does not give very precise recipes for making these parameter choices in branch-and-cut for getting the best performance on particular instances. There are some general insights available from theory, but a large part of the actual deployment of these algorithms in practice is heuristic in nature. This state of affairs has prompted several groups of researchers to explore the possibility of using machine learning tools to make these parameter choices in branch-and-cut; see [Scavuzzo et al., 2024] for a survey and the references therein.

In this paper we focus on some foundational aspects of the use of machine learning to improve algorithmic performance of branch-and-cut. One can formalize the problem as follows. Given a family of instances of integer programming problems, one wishes to select the parameters of branch-and-cut that will perform well on average, and one wishes to do this in a data-driven manner. More formally, one assumes there is a probability distribution over the family of instances and the goal is to find the choice of parameters that gives the best performance in expectation with respect to this distribution. The probability distribution is not explicitly known, but one can sample instances from the distribution and use these samples to guide the choice of parameters. This puts the problem in a classical learning theory framework and one can ask questions like the sample size required to guarantee success with high accuracy and high probability (over the samples). This perspective was pioneered in a recent series of papers [Balcan et al., 2021a,c,b, 2022, 2018] with several important insights and technical contributions. The broader question of selecting a good algorithm from a suite of algorithms for a computational problem, given access to samples from a distribution over the instances of the problem, is generally called *data-driven algorithm design* and has received attention recently; see [Balcan, 2020, Balcan et al., 2021a] and the references therein. There is also a lot of recent activity in the related aspect of *algorithm design with predictions*; see, e.g., [Mitzenmacher and Vassilvitskii, 2022].

This paper contributes to this line of research that analyzes the sample complexity of using machine learning tools for parameter selection in branch-and-cut. In particular, we focus on the *cut selection problem*, which is also a central theme in the papers [Balcan et al., 2021c,b, 2022]. Thus, we wish to learn what choice of cutting planes within the branch-and-cut procedure gives the best performance. Our results differ from this previous work in two main ways.

1. In [Balcan et al., 2021c,b, 2022], the authors focus on the classical cutting plane families of Chvátal-Gomory (CG) and Gomory Mixed-Integer (GMI) cutting planes. These are actually two very special cases of a much more general framework for deriving cutting planes for integer programming problems, which is called *cut generating function theory*. The idea behind cut generating functions goes back to seminal work by [Balas, 1971] and [Gomory and Johnson, 1972a,b] from the 1970s, and there has been a tremendous amount of progress in our understanding of this theory in the past 15 years. The main point is that cut generating functions give a parameterized family of cutting planes to choose from, which vastly generalizes the CG and GMI cuts. We provide sample complexity results for cut selection from this larger family. This requires new understanding of the structure of these cutting planes and their interplay with learning theory tools. While we do rely on some of the ideas from [Balcan et al., 2021c,b, 2022], several new technical and algorithmic challenges need to be overcome.

   The full potential of cut generating functions has not been realized despite sustained efforts in the past decade; see [Poirrier, 2014, Chapter 6] for a nuanced discussion. We believe an important factor behind this is that the cut selection problem for this much larger class is very hard. Thus, while they promise significant gains over traditional ideas like GMI cuts, it has been difficult to utilize them in practice. Using modern machine learning tools to help with the cut selection could be the key to deploying these powerful tools in practice. Thus, we believe our results to be significant in that they are the first of their kind in terms of establishing a rigorous foundation for using machine learning to solve the cut selection problem for cut generating functions.

2. All of the work in [Balcan et al., 2021a,c,b, 2022, 2018] focuses on making a *single* choice of parameters that does well in expectation. However, it can be significantly more beneficial to allow the choice of parameters to *depend on the instance* [Rice, 1976, Gupta and Roughgarden, 2016]. In recent work, the use of neural networks was suggested as a way to map instances to the choice of parameters in data-driven algorithm design [Cheng et al.,

2024], and sample complexity bounds were derived for learning such a neural network. The authors in [Cheng et al., 2024] worked with CG and GMI cuts. Here we extend that analysis to learn neural mappings to more general cut generating functions.

Cut generating functions derive cutting planes by using the data from an optimal simplex tableaux for the linear programming relaxation of the integer programming problem. CG and GMI cutting planes are a special case of such cutting planes where data from a *single* tableaux row is used. Our first result (Theorem 3.2) establishes sample complexity bounds for a parameterized family of cut generating functions that uses information from single rows of the simplex tableaux, but goes beyond CG and GMI cuts. Cut generating function theory also allows the use of information from *multiple rows* of the tableaux, which will naturally result in stronger cutting planes because more information from the problem is used to derive these cuts. Our second result (Theorem 4.4) gives sample complexity results for a parameterized family of cut generating functions that uses information from $k$ rows, for any fixed natural number $k \geq 2$. In Section 5, these sample complexity results are extended to learning neural networks that map instances to cut generating functions, achieving an instance dependent performance.

The choice of the cut generating function families that we study in this paper were dictated by three things: 1) we should be able to derive cutting planes from them in a computationally efficient way, 2) we should be able to derive concrete sample complexity bounds, and 3) we should be able to demonstrate that these new cutting planes are better in practice than classical cuts. Sections 3, 4 and 5 achieve aims 1) and 2). Section 6 gives evidence for 3) by showing that our choice of cut generating functions can indeed improve performance of branch-and-cut, especially with the use of instance dependent cutting planes. Our performance criteria is the overall tree size (number of nodes explored by the solver) for solving the problem. For those familiar with cut generating function theory, we mention that our cut generating functions are all *extreme functions*. This provides some additional theoretical underpinning to the claim that these cutting planes are of good quality.

We begin our formal presentation in Section 2 where we introduce the required concepts, terminology and notation from integer programming and learning theory that are needed to state and prove our results formally, which is done in Sections 3, 4 and 5.

## 2 Formal setup of the problem

Throughout the paper, we will use the following standard notations. The sign function $\mathrm{sgn} : \mathbb{R} \to \{0, 1\}$ is defined by $\mathrm{sgn}(x) = 1$ if $x \geq 0$ and $\mathrm{sgn}(x) = 0$ if $x < 0$. The set $\Delta_d^\tau = \left\{ \mathbf{x} \in \mathbb{R}^d : \mathbf{x}_1, \dots, \mathbf{x}_d \geq \frac{1}{\tau}, \sum_{i=1}^d \mathbf{x}_i = 1 \right\}$ represents a $d$-dimensional simplex, where $\tau \geq 2d$ is some predefined number. For a set of vectors $\{\mathbf{x}^1, \dots, \mathbf{x}^t\} \subseteq \mathbb{R}^d$, we use superscripts to denote vector indices and subscripts to specify the coordinates in a vector; thus, $\mathbf{x}_j^i$ refers to the $j$-th coordinate of vector $\mathbf{x}^i$. The floor and ceiling functions, denoted by $\lfloor \cdot \rfloor$ and $\lceil \cdot \rceil$ respectively, round each component of a vector down or up to the nearest integer. We denote $[\mathbf{x}] = \mathbf{x} - \lfloor \mathbf{x} \rfloor$ for any $\mathbf{x} \in \mathbb{R}^d$, representing the fractional part of each component of the vector.

### 2.1 Integer linear programming background

Given positive integers $m, n$, a pure integer linear programming (ILP) problem in canonical form can be described as:

$$
\begin{aligned}
\max \quad & \mathbf{c}^\mathsf{T} \mathbf{x} \\
\text{s.t.} \quad & A\mathbf{x} \leq \mathbf{b}, \mathbf{x} \geq \mathbf{0}, \mathbf{x} \in \mathbb{Z}^n,
\end{aligned}
\tag{1}
$$

for some $A \in \mathbb{Z}^{m \times n}, \mathbf{b} \in \mathbb{Z}^m$, and $\mathbf{c} \in \mathbb{R}^n$. This problem is represented by the tuple $(A, \mathbf{b}, \mathbf{c})$. In this paper, we consider the set of ILP instances $\mathcal{I}$ such that for any $I = (A, \mathbf{b}, \mathbf{c}) \in \mathcal{I}$, there exists a universal constant $\varrho$ such that $\sup \left\{ \lceil \|\mathbf{x}\|_\infty \rceil : A\mathbf{x} \leq \mathbf{b}, \mathbf{x} \geq \mathbf{0} \right\} \leq \varrho$.

A *cutting plane* for (1) is given by $(\boldsymbol{\alpha}, \beta) \in \mathbb{R}^n \times \mathbb{R}$ such that the inequality $\boldsymbol{\alpha}^\mathsf{T} \mathbf{x} \leq \beta$ is satisfied by all points in the feasible region $\{\mathbf{x} \in \mathbb{Z}^n : A\mathbf{x} \leq \mathbf{b}, \mathbf{x} \geq \mathbf{0}\}$ of (1).

**Cut generating functions.** We now present the technique of cut generating functions to derive cutting planes for (1). Consider the equivalent standard form of (1) after introducing *slack variables*:

$$\widetilde{A}\mathbf{y} = \mathbf{b}, \ \mathbf{y} \in \mathbb{Z}_+^{m+n}, \tag{2}$$

where $\widetilde{A} = [A, I] \in \mathbb{Z}^{m \times (n+m)}$. The simplex tableaux of the above problem with respect to a basis $B \subseteq \{1, \ldots, m+n\}$ with $|B| = m$ is given by

$$\mathbf{y}^B + \widetilde{A}_B^{-1}\widetilde{A}_N\mathbf{y}^N = \widetilde{A}_B^{-1}\mathbf{b}, \ \mathbf{y} \in \mathbb{Z}_+^{m+n}, \tag{3}$$

where $\widetilde{A}_B$ and $\widetilde{A}_N$ represent the submatrices of $\widetilde{A}$ corresponding to the columns indexed by $B$ and $N = \{1, \ldots, m+n\} \setminus B$, respectively. We select any $k$ rows, where $k \in \{1, ..., m\}$, from (3) such that the right-hand side vector is not integral, and write the resulting system as:

$$\mathbf{z} + \sum_{i=1}^n \mathbf{r}^i y_i^N = \mathbf{f}, \ \mathbf{y}^N \in \mathbb{Z}_+^n, \mathbf{z} \in \mathbb{Z}_+^k, \tag{4}$$

where $\mathbf{r}^1, \ldots, \mathbf{r}^n \in \mathbb{R}^k$ and $\mathbf{f} \in \mathbb{R}^k \setminus \mathbb{Z}^k$ are subvectors of $\widetilde{A}_B^{-1}\widetilde{A}_N$ and $\widetilde{A}_B^{-1}\mathbf{b}$, respectively, corresponding to the selected rows. Suppose we have a function $\pi : \mathbb{R}^k \to \mathbb{R}_+$ satisfying the following conditions:

1. $\pi(\mathbf{0}) = 0, \pi(\mathbf{f}) = 1$,

2. subadditivity: $\pi(\mathbf{r} + \mathbf{r}') \leq \pi(\mathbf{r}) + \pi(\mathbf{r}'), \forall \mathbf{r}, \mathbf{r}' \in \mathbb{R}^k$,

3. periodicity: $\pi(\mathbf{r} + \mathbf{w}) = \pi(\mathbf{r}), \forall \mathbf{r} \in \mathbb{R}^k, \mathbf{w} \in \mathbb{Z}^k$,

then for any feasible $\mathbf{y}^N$ and $\mathbf{z}$ we have the following inequality:

$$1 = \pi(\mathbf{f}) = \pi\left(\mathbf{z} + \sum_{i=1}^n \mathbf{r}^i y_i^N\right) = \pi\left(\sum_{i=1}^n \mathbf{r}^i y_i^N\right) \leq \sum_{i=1}^n \pi\left(\mathbf{r}^i y_i^N\right) \leq \sum_{i=1}^n \pi\left(\mathbf{r}^i\right) y_i^N. \tag{5}$$

It's noteworthy that the optimal solution to the relaxed linear programming problem, $\left[\mathbf{y}^B, \mathbf{y}^N\right] = \left[\widetilde{A}_B^{-1}\mathbf{b}, \mathbf{0}\right]$, will always violate this inequality. By substituting out the slack variables using (2), the inequality $\sum_{i=1}^n \pi\left(\mathbf{r}^i\right) y_i^N \geq 1$ becomes a cutting plane $\boldsymbol{\alpha}^\mathsf{T}\mathbf{x} \leq \beta$ for (1); see Lemma A.3.

We present two classical examples of one-dimensional (i.e., $k = 1$) cut generating functions here (as well as their plots in Figure 1a and Figure 1b). A family of one-dimensional cut generating functions we will use in this paper will be presented in Section 3, and a family of $k$-dimensional cut generating functions, for arbitrary $k \geq 1$, will be presented in Section 4.

**Gomory fractional cut [Gomory, 1958]:** Define $\mathsf{CG}_f(r) = \frac{[r]}{[f]}$. Applying this function to the $j$-th row of the simplex tableau (4) with $\mathbf{f}_j \notin \mathbb{Z}$, the valid cut (5) translates to the Gomory fractional cut:

$$\sum_{i=1}^n \frac{[\mathbf{r}_j^i]}{[\mathbf{f}_j]}y_i^N \geq 1 \iff \sum_{i=1}^n \frac{\mathbf{r}_j^i - \lfloor \mathbf{r}_j^i \rfloor}{\mathbf{f}_j - \lfloor \mathbf{f}_j \rfloor}y_i^N \geq 1 \iff \sum_{i=1}^n \left(\mathbf{r}_j^i - \lfloor \mathbf{r}_j^i \rfloor\right) y_i^N \geq \mathbf{f}_j - \lfloor \mathbf{f}_j \rfloor.$$

This cut generating function gives cutting planes that are equivalent to the well-known Chvátal-Gomory (CG) cuts; see [Conforti et al., 2014, Section 5.2.4] for a discussion.

**Gomory's mixed-integer cut [Gomory, 1960]:** The GMI cut function $\mathsf{GMI}_f(r)$ is defined as $\frac{[r]}{[f]}$ when $[r] \leq [f]$, and $\frac{1-[r]}{1-[f]}$ when $[r] > [f]$. Applying to the $j$-th row with $\mathbf{f}_j \notin \mathbb{Z}$, the valid cut (5) translates to the GMI cut:

$$\sum_{i:[\mathbf{r}_j^i] \leq [\mathbf{f}_j]} \frac{[\mathbf{r}_j^i]}{[\mathbf{f}_j]}y_i^N + \sum_{i:[\mathbf{r}_j^i] > [\mathbf{f}_j]} \frac{1 - [\mathbf{r}_j^i]}{1 - [\mathbf{f}_j]}y_i^N \geq 1 \iff \sum_{i:[\mathbf{r}_j^i] \leq [\mathbf{f}_j]} [\mathbf{r}_j^i]y_i^N + \frac{[\mathbf{f}_j]}{1 - [\mathbf{f}_j]} \sum_{i:[\mathbf{r}_j^i] > [\mathbf{f}_j]} \left(1 - [\mathbf{r}_j^i]\right) y_i^N \geq [\mathbf{f}_j].$$

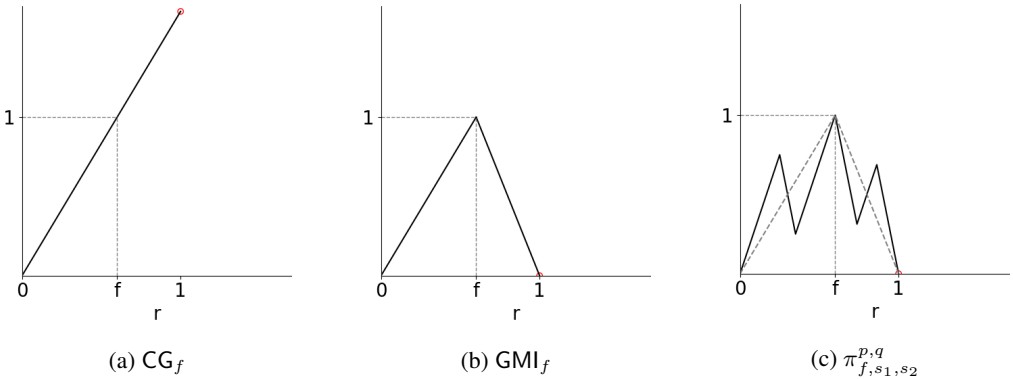

(a) CG$_f$  (b) GMI$_f$  (c) $\pi_{f,s_1,s_2}^{p,q}$

Figure 1: Three cut generating functions on $[0,1)$, where $\pi_{f,s_1,s_2}^{p,q}$ is defined in Section 3.

## 2.2 Sample complexity of selecting cut generating functions

We will consider parameterized families of cut generating functions and the sample complexity of learning which cut generating functions work well. More precisely, we will track how well branch-and-cut performs when the cutting plane corresponding to a specific cut generating function is added to the initial linear programming relaxation of the problem, a.k.a the *root node of the branch-and-cut tree*. In this paper, we consider the branch-and-cut tree built with a product scoring policy for variable selection, a depth-first search policy for node selection, and only one cutting plane added at the root node. We will use the overall tree size needed to solve the problem as the quantitative measure of performance, which is strongly correlated with the overall solve time.

Consider an unknown probability distribution $\mathcal{D}$ over the instance space $\mathcal{I}$. We are presented with problems drawn independently and identically distributed (i.i.d.) from this distribution. We also have a family of cut generating functions parameterized by $\boldsymbol{\mu} \in \mathcal{P}$. Let $c(I, \boldsymbol{\mu})$ denote the cutting plane obtained by applying the cut generating function corresponding to $\boldsymbol{\mu}$ to the instance $I$ as explained in the previous section. $h(I, \boldsymbol{\mu}) \in [0, B] \cap \mathbb{Z}$ will denote the truncated branch-and-cut tree size for some $B > 0$, when the cutting plane $c(I, \boldsymbol{\mu})$ is used at the root node for processing $I$.

The objective is to find the cut generating function that minimizes the expected tree size over the distribution $\mathcal{D}$, i.e., we want to solve the stochastic optimization problem $\min_{\boldsymbol{\mu} \in \mathcal{P}} \mathbb{E}_{I \sim \mathcal{D}} h(I, \boldsymbol{\mu})$. For any $\varepsilon > 0$ and $\delta \in (0, 1)$, the *sample complexity* of the problem is a natural number $N = N(\varepsilon, \delta)$ such that if the number of sampled instances exceeds $N$, the expected tree size for the distribution $\mathcal{D}$ and the average tree size for the sampled instances differ by less than $\varepsilon$ for every $\boldsymbol{\mu} \in \mathcal{P}$, with probability (over the samples) at least $1 - \delta$. Thus, if we have that many samples, we can use the cut generating function that minimizes the average tree size on our instances (this is a deterministic optimization problem since the sample is at hand, a.k.a the *empirical risk minimization (ERM)* or *sample average approximation (SAA)* problem) and we will do well in expectation, i.e., generalize to unseen instances, with high probability.

The pseudo-dimension $\mathrm{Pdim}(\mathcal{H})$ is a measure of the 'complexity' of the associated function class $\mathcal{H} := \{h(\cdot, \boldsymbol{\mu}) : \boldsymbol{\mu} \in \mathcal{P}\}$, and is a key concept closely related to sample complexity. It is defined as the largest integer $t$ for which there exists a set of instances and real values $(I_1, s_1), \ldots, (I_t, s_t) \in \mathcal{I} \times \mathbb{R}$ such that

$$2^t = |\{(\mathrm{sgn}(h(I_1, \boldsymbol{\mu}) - s_1), \ldots, \mathrm{sgn}(h(I_t, \boldsymbol{\mu}) - s_t)) : \boldsymbol{\mu} \in \mathcal{P}\}| \, .$$

A classical result in statistical learning theory (e.g., Theorem 19.2 in [Anthony et al., 1999]) implies the sample complexity bound

$$N(\varepsilon, \delta) = \mathcal{O}\left(\frac{B^2}{\varepsilon^2}\left(\mathrm{Pdim}(\mathcal{H}) \log\left(\frac{B}{\varepsilon}\right) + \log\left(\frac{1}{\delta}\right)\right)\right) \, .$$

Thus, our task reduces to finding an upper bound for $\mathrm{Pdim}(\mathcal{H})$.

In learning theory, identifying the piecewise structure of the function class $\mathcal{H} = \{h(\cdot, \boldsymbol{\mu}) : \boldsymbol{\mu} \in \mathcal{P}\}$ is a standard technique for bounding its pseudo-dimension (see [Anthony et al., 1999, Bartlett et al., 1998, 2019, Sontag et al., 1998, Balcan et al., 2021a]). For a fixed instance $I \in \mathcal{I}$, one shows that

the parameter space $\mathcal{P}$ can be partitioned into regions such that the function $h(I, \boldsymbol{\mu})$ behaves as a fixed 'simple function' within each region. The partition is defined by a set of functions on $\mathcal{P}$ and the regions of the partition correspond to parameter values in $\boldsymbol{\mu} \in \mathcal{P}$ where these functions have invariant signs.

Given that our tree size function is an integer-valued function, we present a particular such result below in Lemma 2.1, which motivates the piecewise structure results presented later in Proposition 3.1 and Proposition 4.3. These results will be used in the proofs of Theorem 3.2 and Theorem 4.4 to bound the pseudo-dimension. The proof of Lemma 2.1 is provided in Appendix A. Note that a more general version of this result is given in [Balcan et al., 2021a], but our specific version is asymptotically better than a direct application of the general result, since Sauer's lemma ([Sauer, 1972, Shelah, 1972]) is not involved in the proof.

**Lemma 2.1.** Let $h : \mathcal{I} \times \mathcal{P} \to \mathbb{Z}$, where $\mathcal{P} \subseteq \mathbb{R}^d$ for a natural number $d$. Suppose that for any fixed $I \in \mathcal{I}$, there exist at most $\Gamma$ functions, each expressible as the quotient of a polynomial of degree at most $a$ and a strictly positive function. These functions partition $\mathcal{P}$ into regions in which $h(I, \boldsymbol{\mu})$ remains constant. Then, the pseudo-dimension of the function class $\{h(\cdot, \boldsymbol{\mu}) : \boldsymbol{\mu} \in \mathcal{P}\}$ is given by:

$$\text{Pdim}\left(\{h(\cdot, \boldsymbol{\mu}) : \boldsymbol{\mu} \in \mathcal{P}\}\right) = \mathcal{O}(d \log(\Gamma a)).$$

**Remark 2.2.** In Theorems 3.2 and 4.4 that bound the pseudo-dimensions, we assume that the row(s) used for generating the cut are prefixed for all instances $I \in \mathcal{I}$. However, it is straightforward to extend the analysis to give bounds on the pseudo-dimension when the choice of rows is also learned along with the cut generating function. We leave these details out of this conference version of the paper.

# 3  One-dimensional cut generating functions

In this section, we present a family of one-dimensional cut generating functions (originally proposed in [Gomory and Johnson, 2003]) that we believe satisfy the three criteria laid out in the Introduction, i.e., cutting planes can be obtained efficiently from them (we present closed form formulas below), sample complexity (pseudo-dimension) bounds can be established rigorously (Theorem 3.2 below), and they result in significantly smaller tree sizes compared to traditional cutting planes (Section 6).

## 3.1  The construction

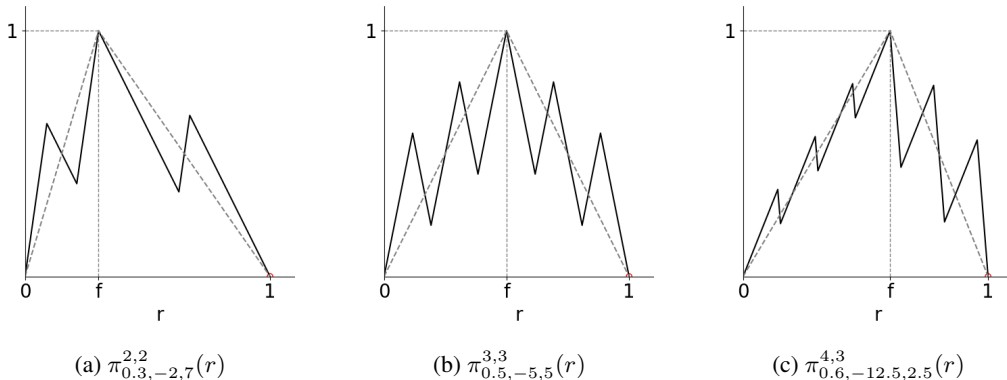

(a) $\pi_{0.3,-2,7}^{2,2}(r)$        (b) $\pi_{0.5,-5,5}^{3,3}(r)$        (c) $\pi_{0.6,-12.5,2.5}^{4,3}(r)$

Figure 2: Three examples of the one-dimensional cut generating functions $\pi_{f,s_1,s_2}^{p,q}$ on $[0,1)$.

For any $f \in (0,1), p, q \in [2, +\infty] \cap \mathbb{Z}, s_1, s_2 \in \mathbb{R}$, let

$$\pi_{f,s_1,s_2}^{p,q}(r) = \max \left\{ \left\{ \min \left\{ \phi_i^1(r), \phi_i^2(r) \right\} : i = 1, \ldots, p \right\} \cup \left\{ \min \left\{ \psi_j^1(r), \psi_j^2(r) \right\} : j = 1, \ldots, q-1 \right\} \right\},$$

where

$$\phi_i^1(r) = s_1 r + i\frac{1 - fs_1}{p}, \quad \phi_i^2(r) = s_2 r + (i-1)\frac{1 - fs_2}{p-1}, i = 1, \ldots, p,$$

$$\psi_j^1(r) = s_1(r-1) + (j-1)\frac{1 + (1-f)s_1}{q-1}, \quad \psi_j^2(r) = s_2(r-1) + j\frac{1 + (1-f)s_2}{q}, j = 1, \ldots, q-1.$$

See Figures 1c and 2 for examples with different $f, p, q$ and $s_1, s_2$. Let $\mathcal{D}_f^{p,q}$ denote the set of all $(s_1, s_2)$ such that $\pi_{f,s_1,s_2}^{p,q}(\cdot)$ is a valid cut generating function, i.e., it satisfies the three conditions outlined in Section 2.1. The closed form of this set is provided in Lemma B.1, which indicates that $\mathcal{D}_f^{p,q}$ is always a (possibly semi-infinite) rectangle.

## 3.2 Pseudo-dimension bound

We first show that the cutting plane coefficients for (2) derived from this family of cut generating functions has a piecewise affine linear structure. This is key to establishing the pseudo-dimension bounds in Theorem 3.2 using Lemma 2.1.

**Proposition 3.1.** For any fixed $f \in (0, 1)$, $p, q \in \mathbb{N} \cap [2, +\infty]$, and $r_1, \ldots, r_n \in [0, 1)$, there exists a decomposition of the $(s_1, s_2)$ space $\mathcal{D}_f^{p,q}$ given by at most $n$ hyperplanes such that, within each region, each coordinate of the cutting plane $\left[ \pi_{f,s_1,s_2}^{p,q}(r_1), \ldots, \pi_{f,s_1,s_2}^{p,q}(r_n) \right]^\mathsf{T}$ is a fixed affine linear function of $(s_1, s_2)$.

To make the parameter selection in more controlled manner, we introduce a large positive constant $M$ and adjust the bounds of $s_1$ and $s_2$ given in Lemma B.1 to finite ranges by replacing $-\infty$ and $+\infty$ with $\frac{1}{f-1} - M$ and $\frac{1}{f} + M$, respectively, in the corresponding cases. This restricts $(s_1, s_2)$ to the product of 2 bounded intervals, denoted as $[l_1, u_1] \times [l_2, u_2]$, which is a bounded subset of $\mathcal{D}_f^{p,q}$. This allows us to use parameters $\boldsymbol{\mu} = (\boldsymbol{\mu}_1, \boldsymbol{\mu}_2) \in [0, 1]^2$ to control $s_1$ and $s_2$ as follows:

$$s_1 = u_1 - \boldsymbol{\mu}_1(u_1 - l_1), \ s_2 = l_2 + \boldsymbol{\mu}_2(u_2 - l_2). \tag{6}$$

We remark that setting $\boldsymbol{\mu}_1 = \boldsymbol{\mu}_2 = 0$ gives the $\mathsf{GMI}_f$ function.

We now have all the pieces to state the precise pseudo-dimension bound.

**Theorem 3.2.** Let $p, q \geq 2$ be arbitrary, but fixed, natural numbers. Let $T(I, \boldsymbol{\mu})$ denote the tree size of the branch-and-cut algorithm after adding the cut induced by the cut generating function $\pi_{f,s_1,s_2}^{p,q}(\cdot)$ at the root for a given instance $I \in \mathcal{I}$, where $s_1, s_2$ are given by the mappings (6) based on $\boldsymbol{\mu}_1, \boldsymbol{\mu}_2$, and $f$ is determined by $I$. Then, the pseudo-dimension is given by

$$\mathrm{Pdim}\left( \{ T(\cdot, \boldsymbol{\mu}) : \mathcal{I} \to [0, B] \mid \boldsymbol{\mu} \in [0, 1]^2 \} \right) = \mathcal{O}\left( n^2 \log((m+n)\varrho) \right).$$

# 4 $k$-dimensional cut generating functions

In this section, we present a family of $k$-dimensional cut generating functions, for arbitrary $k \geq 2$, that we believe satisfy the three criteria laid out in the Introduction, i.e., cutting planes can be obtained efficiently from them (Theorem 4.2 below), sample complexity (pseudo-dimension) bounds can be established rigorously (Theorem 4.4 below), and they result in smaller tree sizes compared to traditional cutting planes (Section 6). We note that this particular family of cut generating functions has not been studied previously in the literature (from a theory or computational perspective), though they are a subclass of cut generating functions studied in [Basu and Sankaranarayanan, 2019].

## 4.1 The construction

Recall that $m$ is the total number of constraints in the original integer program (1), and $k \leq m$ is the number of rows from the simplex tableaux used to derive the cutting planes. For any $\mathbf{f} \in [0, 1)^k \backslash \{\mathbf{0}\}$ and $\boldsymbol{\mu} = [\boldsymbol{\mu}_1, \ldots, \boldsymbol{\mu}_k]^\mathsf{T} \in \Delta_k^\tau$ with a universal large constant $\tau \geq 2m$, let

$$\mathbf{a}^0 = \frac{\sum_{i=1}^k \boldsymbol{\mu}_i \mathbf{e}^i}{\sum_{i=1}^k \boldsymbol{\mu}_i \mathbf{f}_i}, \ \mathbf{a}^1 = \frac{1}{\mathbf{f}_1 - 1}\mathbf{e}^1, ..., \mathbf{a}^k = \frac{1}{\mathbf{f}_k - 1}\mathbf{e}^k \in \mathbb{R}^k,$$

where $\mathbf{e}^i$ denotes the $i$-th standard basis vector in $\mathbb{R}^k$. Define $\pi_{\mathbf{f}, \boldsymbol{\mu}} : \mathbb{R}^k \to \mathbb{R}$ by

$$\pi_{\mathbf{f}, \boldsymbol{\mu}}(\mathbf{r}) := \min_{\mathbf{z} \in \mathbb{Z}^k} \max_{i=0,...,k} \langle \mathbf{a}^i, \mathbf{r} + \mathbf{z} \rangle.$$

Using well-known results from cut generating function theory, it can be shown that $\pi_{\mathbf{f}, \boldsymbol{\mu}}$ satisfies the three conditions in Section 2.1 to qualify as a cut generating function; see, for example, the analysis in [Basu and Sankaranarayanan, 2019]. It is noteworthy that for each $\boldsymbol{\mu} = \mathbf{e}^i$ with $i \in \{1, \ldots, k\}$, the function $\pi_{\mathbf{f}, \boldsymbol{\mu}}$ is equivalent to the one-dimensional GMI function $\mathsf{GMI}_{\mathbf{f}_i}$, defined in Section 2.1.

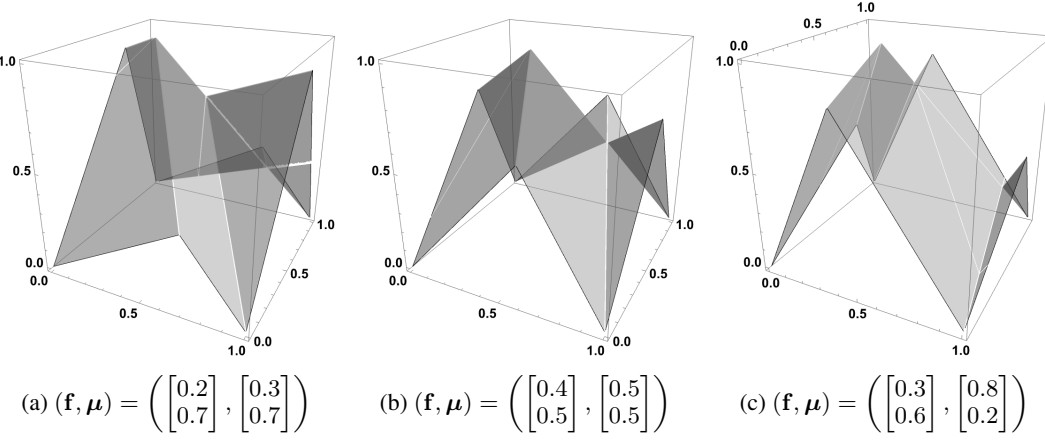

(a) $(\mathbf{f}, \boldsymbol{\mu}) = \left( \begin{bmatrix} 0.2 \\ 0.7 \end{bmatrix}, \begin{bmatrix} 0.3 \\ 0.7 \end{bmatrix} \right)$     (b) $(\mathbf{f}, \boldsymbol{\mu}) = \left( \begin{bmatrix} 0.4 \\ 0.5 \end{bmatrix}, \begin{bmatrix} 0.5 \\ 0.5 \end{bmatrix} \right)$     (c) $(\mathbf{f}, \boldsymbol{\mu}) = \left( \begin{bmatrix} 0.3 \\ 0.6 \end{bmatrix}, \begin{bmatrix} 0.8 \\ 0.2 \end{bmatrix} \right)$

Figure 3: Three examples of the 2-dimensional cut generating functions $\pi_{\mathbf{f}, \boldsymbol{\mu}}$ on $[0,1)^2$.

**Remark 4.1.** As we pointed out in the introduction, both families of cut generating functions considered in this paper are extreme for the pure integer infinite relaxation problem [Conforti et al., 2014, Gomory and Johnson, 1972a,b]. The first family, described in Section 3.1, was proved to be extreme in Gomory and Johnson's original paper [Gomory and Johnson, 2003]. For $k$-dimensional CGFs, they are minimal valid functions as they are the trivial lifting of the gauge function of maximal $(\mathbf{f} + \mathbb{Z}^k)$-free convex sets with the covering property [Basu et al., 2013a, Conforti et al., 2014, Averkov and Basu, 2015]. Then, these $k$-dimensional CGFs are extreme by the $(k+1)$-slope theorem [Basu et al., 2013b].

## 4.2 Computation and pseudo-dimension

Algorithm 1 shows how to compute the function values $\pi_{\mathbf{f}, \boldsymbol{\mu}}(\mathbf{r})$ (the cutting plane coefficients), in time that is linear in the dimension $k$. We then expose an important piecewise structure of the corresponding family of cutting planes for (2) in Proposition 4.3. This piecewise structure is the key to establishing upper bounds on the pseudo-dimension (recall Lemma 2.1) for learning the optimal cut generating function from this family in Theorem 4.4.

---

**Algorithm 1** Computation of $\pi_{\mathbf{f}, \boldsymbol{\mu}}(\mathbf{r})$

---

1: **Input:** $k \in \mathbb{N}_+ \cap [2, \infty), \mathbf{f} \in [0,1)^k \backslash \{\mathbf{0}\}, \boldsymbol{\mu} = [\boldsymbol{\mu}_1, \ldots, \boldsymbol{\mu}_k]^\mathsf{T} \in \Delta_k^\tau, \mathbf{r} \in \mathbb{R}^k$
2: **Output:** $\pi_{\mathbf{f}, \boldsymbol{\mu}}(\mathbf{r})$
3: $\bar{\mathbf{r}} \leftarrow \mathbf{r} - \lfloor \mathbf{r} \rfloor - \sum_{i=1}^k \mathbb{1}(\mathbf{r}_i \geq \mathbf{f}_i + \lfloor \mathbf{r}_i \rfloor) \mathbf{e}^i$        $\triangleright$ $\mathbb{1}(\cdot)$ is the indicator function
4: $p \leftarrow \sum_{i=1}^k \boldsymbol{\mu}_i \bar{\mathbf{r}}_i$
5: $q \leftarrow \sum_{i=1}^k \boldsymbol{\mu}_i \mathbf{f}_i$
6: $\mathbf{s} \leftarrow \left[ \frac{\bar{\mathbf{r}}_1}{\mathbf{f}_1 - 1}, \ldots, \frac{\bar{\mathbf{r}}_k}{\mathbf{f}_k - 1} \right]$
7: $a \leftarrow \max \{ \mathbf{s}_i : i \in \{1, \ldots, k\} \}$
8: $i^* \leftarrow \arg \max \{ \mathbf{s}_i : i \in \{1, \ldots, k\} \}$
9: $b \leftarrow \max \{ \mathbf{s}_i : i \in \{1, \ldots, k\} \backslash \{i^*\} \}$
10: $\lambda^* \leftarrow \frac{\bar{\mathbf{r}}_{i^*} q - (\mathbf{f}_{i^*} - 1)p}{\boldsymbol{\mu}_{i^*}(\mathbf{f}_{i^*} - 1) - q}$
11: $\pi_{\mathbf{f}, \boldsymbol{\mu}}(\mathbf{r}) \leftarrow \min \left\{ \max \left\{ \frac{p + \boldsymbol{\mu}_{i^*} \lceil \lambda^* \rceil}{q}, \frac{\bar{\mathbf{r}}_{i^*} + \lceil \lambda^* \rceil}{\mathbf{f}_{i^*} - 1}, b \right\}, \max \left\{ \frac{p + \boldsymbol{\mu}_{i^*} \lfloor \lambda^* \rfloor}{q}, \frac{\bar{\mathbf{r}}_{i^*} + \lfloor \lambda^* \rfloor}{\mathbf{f}_{i^*} - 1}, b \right\}, \max \left\{ \frac{p}{q}, a \right\} \right\}$

---

**Theorem 4.2.** For any $\mathbf{f} \in [0,1)^k \backslash \{\mathbf{0}\}$, $\tau \geq 2k$, and $\boldsymbol{\mu} \in \Delta_k^\tau$, Algorithm 1 computes $\pi_{\mathbf{f}, \boldsymbol{\mu}}(\mathbf{r})$ in $\mathcal{O}(k)$ time. Therefore, the cutting plane obtained from $\pi_{\mathbf{f}, \boldsymbol{\mu}}$ can be computed in $\mathcal{O}(kn)$ time.

**Proposition 4.3.** For any fixed $k \in \mathbb{Z} \cap [2, +\infty], \mathbf{f} \in [0,1)^k \backslash \{\mathbf{0}\}, \mathbf{r}^1, \ldots, \mathbf{r}^n \in \mathbb{R}^k$, and $\tau \geq 2k$, there exists a decomposition of $\Delta_k^\tau$ obtained by at most $2n(\tau + 3)^2$ hyperplanes such that within each region, each coordinate of the cutting plane $\left[ \pi_{\mathbf{f}, \boldsymbol{\mu}}(\mathbf{r}^1), \ldots, \pi_{\mathbf{f}, \boldsymbol{\mu}}(\mathbf{r}^n) \right]^\mathsf{T}$ is a fixed rational function given by the quotient of two affine linear functions of $\boldsymbol{\mu}$, where the denominator is always a fixed positive function of $\boldsymbol{\mu}$.

**Theorem 4.4.** For any fixed $k \in \mathbb{Z} \cap [2, +\infty]$, let $T^k(I, \boldsymbol{\mu})$ denote the tree size of the branch-and-cut algorithm after adding the cut induced by $\pi_{\mathbf{f}, \boldsymbol{\mu}}$ at the root for a given instance $I \in \mathcal{I}$, where $\mathbf{f}$ is determined by $I$. Then, the pseudo-dimension is given by

$$\mathrm{Pdim}\left(\left\{T^k(\cdot, \boldsymbol{\mu}) : \mathcal{I} \to [0, B] \mid \boldsymbol{\mu} \in \Delta_k^\tau\right\}\right) = \mathcal{O}\left(kn^2 \log((m+n)\varrho) + k^2 \log(n\tau)\right).$$

## 5 Learnability of instance-dependent cut generating functions

The authors in [Cheng et al., 2024] studied the learnability of neural networks that select instance-dependent algorithms for any computational problem, as opposed to selecting a single algorithm that has the best expected performance, and applied this to the problem of selecting from the Chvatal-Gomory cutting plane family. Inspired by their work, this section discusses employing neural networks to dynamically select the most suitable cut generating function from a given family, tailored to each instance, as opposed to selecting a single cut generating function that has the best expected performance overall (as was done in Sections 3 and 4). In other words, given access to samples from the instance distribution, we want to learn the parameters of the optimal neural network that will map instances to instance specific cut generating functions.

We define a neural network $\varphi_\ell : \mathbb{R}^d \times \mathbb{R}^W \to \mathbb{R}^\ell$ consisting of a fully connected architecture with ReLU activations, $L$ layers, $W$ parameters, $d$ input units, $\ell$ output units, and $U$ units in total. An encoder function $\mathrm{Enc} : \mathcal{I} \to \mathbb{R}^d$ is employed to transform instances $I = (A, \mathbf{b}, \mathbf{c}) \in \mathcal{I}$ into suitable neural network inputs, where a straightforward choice for $\mathrm{Enc}$ could be the flattening of $I$ into a vector, although more complex encoding strategies can be adopted to capture additional structural information about the instances. The primary goal is to input the encoded instance $\mathrm{Enc}(I)$ into the neural network, which then predicts parameters for the cut generating function that were discussed in Section 3 and Section 4. This idea is supported by empirical evidence of performance improvements when enumerating cutting plane parameters in an instance-dependent manner, as demonstrated in Table 1. A direct application of the main theorem in [Cheng et al., 2024] yields the following results:

**Theorem 5.1.** Let $h, h^k : \mathcal{I} \times \mathbb{R}^W \to [0, B]$ denote the branch-and-cut tree size after adding the cutting planes induced by corresponding cut generating functions, using parameters determined by the neural network described above. Formally, they are defined as $h(I, \mathbf{w}) = T(I, \varphi_2(\mathrm{Enc}(I), \mathbf{w}))$ and $h^k(I, \mathbf{w}) = T^k(I, \varphi_k(\mathrm{Enc}(I), \mathbf{w}))$, where $T$ and $T^k$ are the tree size functions defined in Theorems 3.2 and 4.4 respectively. Then the pseudo-dimension of these two learning problems have the following upper bounds:

$$\mathrm{Pdim}\left(\left\{h(\cdot, \mathbf{w}) : \mathbf{w} \in \mathbb{R}^W\right\}\right) = \mathcal{O}\left(LW \log(U + 2) + n^2 W \log((m+n)\varrho)\right),$$

$$\mathrm{Pdim}\left(\left\{h^k(\cdot, \mathbf{w}) : \mathbf{w} \in \mathbb{R}^W\right\}\right) = \mathcal{O}\left(LW \log(U + k) + kW \log(n\tau) + n^2 W \log((m+n)\varrho)\right).$$

## 6 Numerical experiments

**Setup.** We conducted numerical experiments to evaluate the performance of both one-dimensional and $k$-dimensional cut generating functions, as discussed in Section 3 and Section 4, across various distributions. The performance of these functions was compared to the GMI cut. The parameters for the cut generating functions were selected to minimize the average branch-and-cut tree size on the training set of size 100, and these parameters were then applied to the test set of size 100 to evaluate performance. All results presented in Table 1 are based on the test set, except for the last column. The experiments were run on a Linux machine equipped with a 12-core Intel i7-12700F CPU and 32GB of RAM. We solved the integer programming problems using Gurobi 11.0.1 [Gurobi Optimization, LLC, 2023], with default cuts, heuristics, and presolve settings turned off. The code and data used in all experiments are available at `https://github.com/Hongyu-Cheng/LearnCGF`.

**Problem descriptions.** We considered two types of problems: Knapsack and Packing.

1. Knapsack$(N, K)$: Multiple knapsack problem with $N$ items and $K$ knapsacks. Instances were generated using the so-called "Chvátal distribution" from [Balcan et al., 2021b]. Note that there are some trivial upper bound constraints on the variables that contribute to the simplex tableaux. Even so, the table has n/a entries for the $k$-row cut strategies for the 1-knapsack problem, as there are not enough fractional rows to generate multi-row cuts for many instances.

2. Packing$(m, n)$: Packing problem with $m$ constraints and $n$ variables. We note that the knapsack problem can be viewed as a special case of the packing problem with binary variables. The packing instances were generated using the distribution from [Tang et al., 2020].

**Cutting plane strategies.** The following cutting plane strategies were evaluated and compared:

1. GMI: Classical Gomory's mixed integer cut as defined in Section 2.1.

2. 1-row cut: Generated using the one-dimensional cut generating function defined in Section 3. We fixed $p = q = 2$ and performed a grid search with a step size of 0.1 to select the best parameter $\boldsymbol{\mu} \in \{0, 0.1, \ldots, 0.9, 1\}^2$.

3. $k$-row cut: Generated using $\pi_{\mathbf{f}, \boldsymbol{\mu}}$ defined in Section 4 with $k \in \{2, 5, 10\}$. We uniformly sampled 121 different $\boldsymbol{\mu}$ on the simplex $\Delta_k^\infty$ (see [Gordon-Rodriguez et al., 2020, Smith and Tromble, 2004]), and selected the best parameter based on the training set.

4. Best 1-row cut: The average tree size using the best parameter for each instance on the *training set*. This is not a practical strategy but indicates the strength of the cut generating functions and the potential for instance-dependent cut generating using neural networks.

Since we aim to demonstrate that cut generating functions can produce stronger cuts than classical cutting planes, we did not specifically consider which row to select to generate the cut. This problem, while important in integer programming literature, is outside the scope of this paper. All cuts are generated from the first row of the simplex tableau with a non-integer right-hand side, and the $k$-row cut is generated from the first $k$ such rows. Also, to select the best parameters based on samples (i.e., solve the ERM problem), we used a simple grid search and optimized through enumeration, since the branch-and-cut tree size is a highly sophisticated function of the added cutting planes at the root node.

**Numerical results.** As shown in Table 1, the two cut generating function families considered in this paper reduce the size of the branch-and-cut tree compared to the GMI cut. Although the improvement over the GMI cut on the test set is less obvious for the packing problem, the last column in the table shows that there are still cutting planes that perform much better for each instance. Moreover, all the multi-row cuts on the Knapsack$(30, 3)$ problems outperform the best 1-row cut, indicating that multi-row cuts can sometimes achieve performance levels that single-row cuts cannot reach.

Table 1: Average tree sizes on 100 instances, after adding a single type of cut at the root.

| **Problem Type** | GMI | 1-row cut | 2-row cut | 5-row cut | 10-row cut | best 1-row cut |
|---|---|---|---|---|---|---|
| Knapsack$(20, 1)$ | 158.88 | **87.0** | n/a | n/a | n/a | 35.54 |
| Knapsack$(30, 1)$ | 832.16 | **58.84** | n/a | n/a | n/a | 13.98 |
| Knapsack$(50, 1)$ | 3543.91 | **277.01** | n/a | n/a | n/a | 125.85 |
| Knapsack$(16, 2)$ | 399.86 | 316.8 | **178.68** | 234.09 | 203.66 | 102.07 |
| Knapsack$(30, 3)$ | 4963.91 | 4311.04 | 3430.37 | **2817.55** | 2822.37 | 3561.36 |
| Packing$(15, 30)$ | 389.67 | **367.48** | 376.86 | 401.87 | 391.44 | 303.72 |
| Packing$(20, 40)$ | 1200.55 | 1123.9 | 1214.92 | **1113.82** | 1185.26 | 738.58 |

## Acknowledgments and Disclosure of Funding

Both authors gratefully acknowledge support from Air Force Office of Scientific Research (AFOSR) grant FA95502010341 and National Science Foundation (NSF) grant CCF2006587. The first author also acknowledges support from the Johns Hopkins University Mathematical Institute for Data Science (MINDS) Fellowship, the Duncan Award 24-33, and the Rufus P. Isaacs Graduate Fellowship.

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

# A Auxiliary Lemmas

**Lemma A.1** (Theorem 5.5 in [Matousek, 1999], Lemma 17 in [Bartlett et al., 2019], Lemma 3.3 in [Anthony et al., 1999], Proposition 2.4 in [Stanley et al., 2004]). *Let $\phi_1, \ldots, \phi_t : \mathbb{R}^d \to \mathbb{R}$ be $t$ multivariate polynomials of degree at most $a$ with $t \geq d$ and $a \geq 1$. Then*

$$|\{(\mathrm{sgn}(\phi_1(\boldsymbol{\mu})), \ldots, \mathrm{sgn}(\phi_t(\boldsymbol{\mu}))) : \boldsymbol{\mu} \in \mathbb{R}^d\}| \leq \begin{cases} \left(\frac{et}{d}\right)^d, & \text{if } a = 1, \\ 2\left(\frac{2eta}{d}\right)^d, & \text{if } a \geq 2, \end{cases}$$

*where $e$ is the base of the natural logarithm.*

*Proof of Lemma 2.1.* For any $t \in \mathbb{N}_+ \cap [d, +\infty)$ and $(I_1, s_1), \ldots, (I_t, s_t) \in \mathcal{I} \times \mathbb{R}$, there are at most $\Gamma t$ rational functions, denote as $\frac{p_1}{q_1}, \ldots, \frac{p_{\Gamma t}}{q_{\Gamma t}}$, where each $p_i$ is a multivariate polynomial of degree at most $a$ on $\boldsymbol{\mu} \in \mathcal{P} \subseteq \mathbb{R}^d$, and each $q_i > 0$ on $\mathcal{P}$. For any $\boldsymbol{\mu}$ within each of these decomposed regions, the vector $[h(I_1, \boldsymbol{\mu}), \ldots, h(I_t, \boldsymbol{\mu})]^\mathsf{T}$ is invariant. The number of the decomposed regions is given by

$$\left| \left\{ \left( \mathrm{sgn}\left(\frac{p_1(\boldsymbol{\mu})}{q_1(\boldsymbol{\mu})}\right), \ldots, \mathrm{sgn}\left(\frac{p_{\Gamma t}(\boldsymbol{\mu})}{q_{\Gamma t}(\boldsymbol{\mu})}\right) \right) : \boldsymbol{\mu} \in \mathcal{P} \right\} \right|$$

$$\leq \left| \left\{ \left( \mathrm{sgn}\left(p_1(\boldsymbol{\mu})\right) \right), \ldots, \mathrm{sgn}\left(p_{\Gamma t}(\boldsymbol{\mu})\right) \right) : \boldsymbol{\mu} \in \mathbb{R}^d \right\} \right|$$

$$\leq 2\left(\frac{2e\Gamma ta}{d}\right)^d,$$

where the last inequality holds by Lemma A.1. We denote these regions as $Q_1, \ldots, Q_{\widetilde{K}}$, where $\widetilde{K} \leq 2\left(\frac{2e\Gamma ta}{d}\right)^d$. Then we have,

$$|\{(\mathrm{sgn}(h(I_1, \boldsymbol{\mu}) - s_1), \ldots, \mathrm{sgn}(h(I_t, \boldsymbol{\mu}) - s_t)) : \boldsymbol{\mu} \in \mathcal{P}\}|$$

$$\leq \sum_{i=1}^{\widetilde{K}} |\{(\mathrm{sgn}(h(I_1, \boldsymbol{\mu}) - s_1), \ldots, \mathrm{sgn}(h(I_t, \boldsymbol{\mu}) - s_t)) : \boldsymbol{\mu} \in Q_i\}|$$

$$= \sum_{i=1}^{\widetilde{K}} 1$$

$$\leq 2\left(\frac{2e\Gamma ta}{d}\right)^d,$$

where the equality holds since each $h(I_j, \boldsymbol{\mu}) - s_j$ is an invariant constant for $\boldsymbol{\mu}$ varying in any fixed $Q_i$. Therefore, the pseudo-dimension is the largest $t$ such that

$$2^t \leq 2\left(\frac{2e\Gamma ta}{d}\right)^d,$$

which is bounded by the largest $t$ such that

$$\frac{1}{2}(t-1) \leq d\log\left(\frac{2e\Gamma ta}{d}\right) \leq d\left(\frac{2e\Gamma ta/d}{8e\Gamma a} + \log\left(8\Gamma a\right)\right) = \frac{1}{4}t + d\log\left(8\Gamma a\right),$$

where the second inequality holds because $\log(x) \leq \frac{x}{c} + \log\left(\frac{c}{e}\right)$ for positive reals $x$ and $c$. Then it follows that $\mathrm{Pdim}(\mathcal{H}) = O(d\log(\Gamma a))$. $\qquad\square$

**Lemma A.2** (Theorem 4.5 in Balcan et al. [2022]). *For any fixed $I = (A, \mathbf{b}, \mathbf{c}) \in \mathcal{I}$, there are at most $\mathcal{O}\left((14)^n(m+2n)^{3n^2}\varrho^{5n^2}\right)$ degree 5 polynomials decomposing the cutting plane space $\mathbb{R}^{n+1}$ into regions such that the size of the branch-and-cut tree after adding the cut $\boldsymbol{\alpha}^\mathsf{T}\mathbf{x} \leq \beta$ at the root remain the same over all $(\boldsymbol{\alpha}, \beta)$ within a given decomposed region.*

**Lemma A.3** ([Tang et al., 2020, Conforti et al., 2014]). *For every $A \in \mathbb{Z}^{m \times n}, \mathbf{b} \in \mathbb{Z}^m$, there exists an affine linear transformation that maps a cutting plane derived from a cut generating function for the standard form integer linear programming feasible region $\{(\mathbf{x}, \mathbf{s}) \in \mathbb{R}^n \times \mathbb{R}^m : A\mathbf{x} + \mathbf{s} = \mathbf{b}, \mathbf{x}, \mathbf{s} \geq 0, \mathbf{x} \in \mathbb{Z}^n, \mathbf{s} \in \mathbb{Z}^m\}$ into the corresponding cutting plane for the corresponding canonical form feasible region $\{\mathbf{x} \in \mathbb{R}^n : A\mathbf{x} \leq \mathbf{b}, \mathbf{x} \geq 0, \mathbf{x} \in \mathbb{Z}^n\}$.*

*Proof.* Let $\mathbf{a_x^T x} + \mathbf{a_s^T s} \geq 1$ be a cutting plane for the standard form integer linear programming problem. For any feasible $\mathbf{x}$ and $\mathbf{s}$, we have $\mathbf{s} = \mathbf{b} - A\mathbf{x}$. Substituting into the inequality, we obtain the equivalent cutting plane $\left(\mathbf{a_s^T} A - \mathbf{a_x^T}\right)\mathbf{x} \leq \mathbf{a_s^T b} - 1$ in the canonical form problem. Then it's clear that

$$\begin{bmatrix} \boldsymbol{\alpha}(\mathbf{a}) \\ \beta(\mathbf{a}) \end{bmatrix} = \begin{bmatrix} A^\mathsf{T}\mathbf{a_s} - \mathbf{a_x} \\ \mathbf{b^T a_s} - 1 \end{bmatrix} = \begin{bmatrix} -I & A^\mathsf{T} \\ \mathbf{0} & \mathbf{b^T} \end{bmatrix} \begin{bmatrix} \mathbf{a_x} \\ \mathbf{a_s} \end{bmatrix} - \begin{bmatrix} \mathbf{0} \\ 1 \end{bmatrix}$$

is affine linear on $\mathbf{a} = \begin{bmatrix} \mathbf{a_x} \\ \mathbf{a_s} \end{bmatrix}$ since $A, \mathbf{b}$ are considered to be fixed. $\qquad\square$

## B  Proofs for results in Section 3

Theorem 6 in [Gomory and Johnson, 2003] yields the following lemma.

**Lemma B.1.** Consider fixed $f \in (0,1)$ and $p, q \in [2, +\infty) \cap \mathbb{N}$. $\pi_{f,s_1,s_2}^{p,q}(\cdot)$ is a valid cut generating function, i.e., it satisfies the three conditions outlined in Section 2.1, if $s_1$ and $s_2$ are chosen from $\mathcal{D}_f^{p,q}$, which is established as follows:

$$\mathcal{D}_f^{p,q} = \begin{cases} \left[\frac{p+q-1}{(p+q-1)f-p}, \frac{1}{f-1}\right] \times \left[\frac{1}{f}, \frac{p+q-1}{q-(p+q-1)(1-f)}\right] & \text{if } (p+q-1)f - p < 0 \text{ and } (p+q-1)(1-f) - q < 0, \\ \left[\frac{p+q-1}{(p+q-1)f-p}, \frac{1}{f-1}\right] \times \left[\frac{1}{f}, +\infty\right) & \text{if } (p+q-1)f - p < 0 \text{ and } (p+q-1)(1-f) - q \geq 0, \\ \left(-\infty, \frac{1}{f-1}\right] \times \left[\frac{1}{f}, \frac{p+q-1}{q-(p+q-1)(1-f)}\right] & \text{if } (p+q-1)f - p \geq 0 \text{ and } (p+q-1)(1-f) - q < 0, \\ \left(-\infty, \frac{1}{f-1}\right] \times \left[\frac{1}{f}, +\infty\right) & \text{if } (p+q-1)f - p \geq 0 \text{ and } (p+q-1)(1-f) - q \geq 0. \end{cases}$$

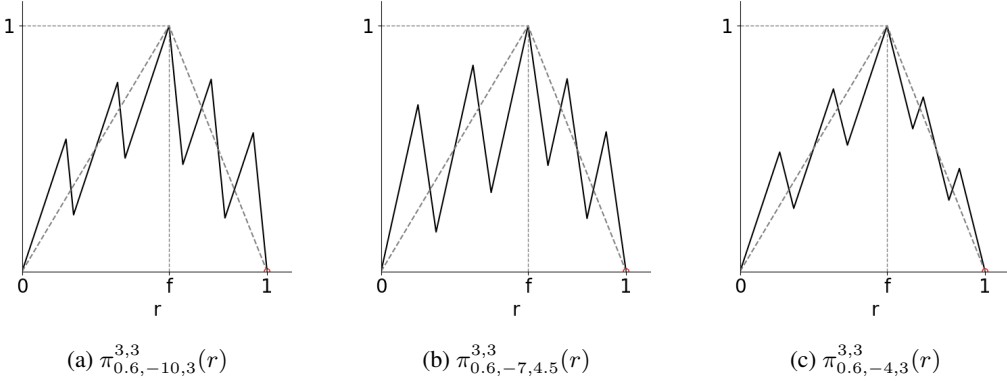

(a) $\pi_{0.6,-10,3}^{3,3}(r)$        (b) $\pi_{0.6,-7,4.5}^{3,3}(r)$        (c) $\pi_{0.6,-4,3}^{3,3}(r)$

Figure 4: For fixed $f \in (0,1), p, q \in [2, +\infty) \cap \mathbb{N}$, the intersection points of $\pi_{f,s_1,s_2}^{p,q}(\cdot)$ and $\mathsf{GMI}_f(\cdot)$ are fixed.

**Lemma B.2.** For any fixed $f \in (0,1), p, q \in [2, +\infty) \cap \mathbb{N}$, and any $(s_1, s_2) \in \mathcal{D}_f^{p,q}$, the intersection points of $\pi_{f,s_1,s_2}^{p,q}(\cdot)$ and $\mathsf{GMI}_f(\cdot)$ are fixed. More specifically, for any $(s_1, s_2) \in \mathcal{D}_f^{p,q}$ with $s_1 \neq \frac{1}{f-1}$ and $s_2 \neq \frac{1}{f}$, the set of intersection points is explicitly given by

$$\left\{ \left(\frac{if}{p}, \frac{i}{p}\right) : i = 0, \ldots, p \right\} \cup \left\{ \left(1 - \frac{j(1-f)}{q}, \frac{j}{q}\right) : j = 0, \ldots, q-1 \right\}$$
$$\cup \left\{ \left(\frac{if}{p-1}, \frac{i}{p-1}\right) : i = 1, \ldots, p-2 \right\} \cup \left\{ \left(1 - \frac{j(1-f)}{q-1}, \frac{j}{q-1}\right) : j = 1, \ldots, q-2 \right\},$$

and these intersection points decompose the interval $[0,1]$ into $2(p+q-2)$ subintervals given by the following break points in ascending order:

$$0, \frac{f}{p}, \frac{f}{p-1}, \ldots, \frac{(p-2)f}{p}, \frac{(p-2)f}{p-1}, \frac{(p-1)f}{p}, f, 1 - \frac{(q-1)(1-f)}{q}, \ldots, 1.$$

*Proof of Lemma B.2.* Let $s_1 < \frac{1}{f-1}$ and $s_2 > \frac{1}{f}$ be any valid slopes. A direct calculation shows that the intersection points of $\pi^{p,q}_{f,s_1,s_2}(r)$ and $\mathsf{GMI}_f(r)$ within $r \in [0,1]$ are given by:

$$\phi^1_i(r) = \mathsf{GMI}_f(r) \iff s_1 r + i\frac{1-fs_1}{p} = \frac{r}{f} \iff r = i\frac{f}{p},$$

$$\phi^2_i(r) = \mathsf{GMI}_f(r) \iff s_2 r + (i-1)\frac{1-fs_2}{p-1} = \frac{r}{f} \iff r = (i-1)\frac{f}{p-1},$$

$$\psi^1_j(r) = \mathsf{GMI}_f(r) \iff s_1(r-1) + (j-1)\frac{1+(1-f)s_1}{q-1} = \frac{1-r}{1-f} \iff r = 1 - (j-1)\frac{1-f}{q-1},$$

$$\psi^2_j(r) = \mathsf{GMI}_f(r) \iff s_2(r-1) + j\frac{1+(1-f)s_2}{q} = \frac{1-r}{1-f} \iff r = 1 - j\frac{1-f}{q},$$

where $i \in \{1,\ldots,p\}$ and $j \in \{1,\ldots,q-1\}$. Eliminating duplicate points and observing a maximum of $2p + 2q - 3$ intersection points in nondegenerate cases, the lemma statement regarding the intersection points is confirmed. The interval decomposition is easy to be verified by sorting these points along their first coordinates. $\qquad\square$

*Proof of Proposition 3.1.* Given any fixed $r_1, \ldots, r_n \in [0,1)$, by Lemma B.2, each $r_j$, $j = 1, \ldots, n$, must lie in one of the intervals independent of $s_1, s_2$ (if $r_j$ is on the boundary of any of these intervals, then $\pi^{p,q}_{f,s_1,s_2}(r_j)$ is a constant). It is not hard to see from the construction and Lemma B.2 that within each interval's interior, there is exactly one nondifferentiable point of $\pi^{p,q}_{f,s_1,s_2}(\cdot)$ in the form of the quotient of two affine linear functions of $s_1$ and $s_2$. For instance,

$$\phi^1_i(r) = \phi^2_i(r) \iff s_1 r + i\frac{1-fs_1}{p} = s_2 r + (i-1)\frac{1-fs_2}{p-1} \iff r = \frac{(i-1)\frac{1-fs_2}{p-1} - i\frac{1-fs_1}{p}}{s_1 - s_2}.$$

We denote these corresponding nondifferentiable points in the form $\frac{f_1(s_1,s_2)}{g_1(s_1,s_2)}, \ldots, \frac{f_n(s_1,s_2)}{g_n(s_1,s_2)}$, where $f_i$ and $g_i$ are affine linear functions of $s_1$ and $s_2$. We introduce the following $n$ hyperplanes of $s_1$ and $s_2$:

$$\{(s_1, s_2) \in \mathcal{D}^{p,q}_f : f_i(s_1, s_2) = r_i g_i(s_1, s_2)\}, \quad i = 1, \ldots, n.$$

Then, within each region decomposed by these hyperplanes, each $\pi^{p,q}_{f,s_1,s_2}(r_j)$ is a fixed affine linear function of $(s_1, s_2)$. $\qquad\square$

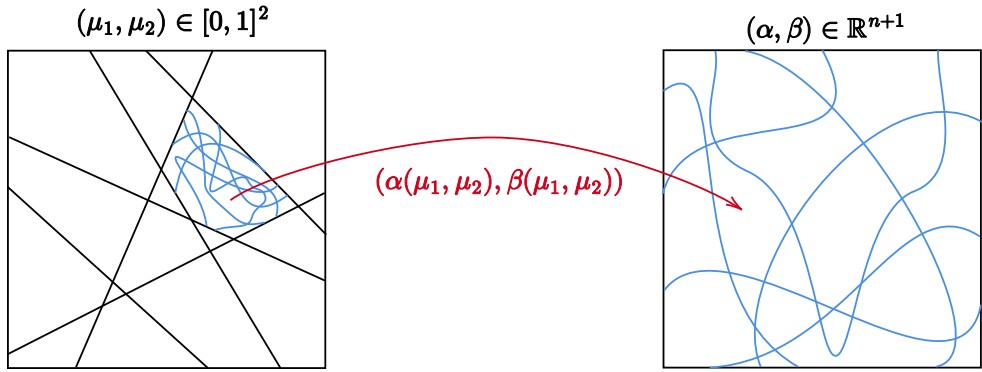

Figure 5: Illustration of the proof of Theorem 3.2.

*Proof of Theorem 3.2.* Lemma A.2 states that there are at most $\Gamma = \mathcal{O}\left((14)^n(m+2n)^{3n^2}\varrho^{5n^2}\right)$ degree 5 polynomials decomposing the cutting plane $(\alpha, \beta)$ space $\mathbb{R}^{n+1}$ such that the branch-and-cut tree size after adding the corresponding cutting plane at the root is the same within each decomposed region. We denote these degree 5 polynomials as $\xi_1, \ldots, \xi_\Gamma$. In other words, for any $(\alpha, \beta) \in \mathbb{R}^{n+1}$ such that the vector

$$\left(\operatorname{sgn}\left(\xi_1\left(\begin{bmatrix}\alpha\\\beta\end{bmatrix}\right)\right), \ldots, \operatorname{sgn}\left(\xi_\Gamma\left(\begin{bmatrix}\alpha\\\beta\end{bmatrix}\right)\right)\right)$$

remains constant, the branch-and-cut tree size after adding $\boldsymbol{\alpha}^\mathsf{T}\mathbf{x} \leq \beta$ at the root is invariant.

Proposition 3.1 provides a decomposition of the $(s_1, s_2)$ space $\mathcal{D}_f^{p,q}$ by $n$ hyperplanes. Consequently, there exists a corresponding decomposition of the $\boldsymbol{\mu}$ space $[0,1]^2$ by $n$ hyperplanes as well, since $\boldsymbol{\mu}$ is a fixed affine linear function of $(s_1, s_2)$ as specified by the mapping (6). We denote these decomposed regions by $Q_1, \ldots, Q_K \subseteq [0,1]^2$, where $K = \mathcal{O}(n^2)$ due to Lemma A.1. Fix a region $Q_i$ for any $i \in \{1, \ldots, K\}$, the mapping (6), Proposition 3.1 and Lemma A.3 give three different fixed affine transformations from $\boldsymbol{\mu}$ to a cutting plane for the canonical linear programming problem. We denote the corresponding cutting plane as

$$\begin{bmatrix} \boldsymbol{\alpha}(\boldsymbol{\mu}) \\ \beta(\boldsymbol{\mu}) \end{bmatrix} = \mathbf{g}(\boldsymbol{\mu}) \in \mathbb{R}^{n+1},$$

where $\mathbf{g}_i, i \in \{1, \ldots, n+1\}$, are affine linear functions of $\boldsymbol{\mu}$. Notice that

$$\left( \operatorname{sgn}\left( \xi_1\left( \begin{bmatrix} \boldsymbol{\alpha}(\boldsymbol{\mu}) \\ \beta(\boldsymbol{\mu}) \end{bmatrix} \right) \right), \ldots, \operatorname{sgn}\left( \xi_\Gamma\left( \begin{bmatrix} \boldsymbol{\alpha}(\boldsymbol{\mu}) \\ \beta(\boldsymbol{\mu}) \end{bmatrix} \right) \right) \right) = \left( \operatorname{sgn}\left( \xi_1(\mathbf{g}(\boldsymbol{\mu})) \right), \ldots, \operatorname{sgn}\left( \xi_\Gamma(\mathbf{g}(\boldsymbol{\mu})) \right) \right),$$

then there are $\Gamma$ degree-5 polynomials

$$(\xi_1 \circ \mathbf{g}), \ldots, (\xi_\Gamma \circ \mathbf{g})$$

over $[0,1]^2$ such that when they have invariant sign patterns, the branch-and-cut tree size after adding the corresponding cutting plane at the root is the same. Thus, there are $\mathcal{O}(K\Gamma)$ such degree 5 polynomials in total. Then the pseudo-dimension result follows from Lemma 2.1. $\qquad\square$

## C   Proofs for results in Section 4

*Proof of Theorem 4.2.* The $\mathcal{O}(k)$ and $\mathcal{O}(nk)$ complexity of Algorithm 1 is straightforward to verify. We now prove the correctness of the algorithm. Given the vectors $\mathbf{a}^0, \mathbf{a}^1, \ldots, \mathbf{a}^k \in \mathbb{R}^k$ defined in Section 4.1, notice that the set

$$\left\{ \mathbf{x} \in \mathbb{R}^k : \langle \mathbf{a}^0, \mathbf{x} \rangle \leq 1, \langle \mathbf{a}^1, \mathbf{x} \rangle \leq 1, \ldots, \langle \mathbf{a}^k, \mathbf{x} \rangle \leq 1 \right\},$$

$$= (\mathbf{f} - \mathbf{1}) + \left\{ \mathbf{x} \in \mathbb{R}^k : \sum_{i=1}^k \boldsymbol{\mu}_i \mathbf{x}_i \leq 1, \mathbf{x}_1 \geq 0, \ldots, \mathbf{x}_k \geq 0 \right\}$$

$$:= G(\mathbf{f}, \boldsymbol{\mu})$$

is a translation of a $k$-dimensional simplex. From the literature on cut generating functions, there exists a so-called lifting region (refer to [Dey and Wolsey, 2010, Basu and Paat, 2015, Basu et al., 2013a]), $R \subseteq G(\mathbf{f}, \boldsymbol{\mu})$, such that for any $\mathbf{r} \in \mathbb{R}^k$, there exists a $\widetilde{\mathbf{r}} \in R \cap (\mathbf{f} + \mathbb{Z}^k)$ such that $\pi_{\mathbf{f},\boldsymbol{\mu}}(\mathbf{r}) = \max_{j=0,\ldots,k} \langle \mathbf{a}^i, \widetilde{\mathbf{r}} \rangle$.

Theorem 2.2 in [Basu and Sankaranarayanan, 2019] implies that

$$R \subseteq G(\mathbf{f}, \boldsymbol{\mu}) \subseteq \cup_{i=1}^k \left( [\mathbf{f}_1 - 1, \mathbf{f}_1] \times \cdots \times [\mathbf{f}_k - 1, \mathbf{f}_k] + \{\lambda \mathbf{e}^i : \lambda \in \mathbb{Z}_+\} \right)$$

$$\subseteq \cup_{i=1}^k \left( [\mathbf{f}_1 - 1, \mathbf{f}_1] \times \cdots \times [\mathbf{f}_k - 1, \mathbf{f}_k] + \{\lambda \mathbf{e}^i : \lambda \in \mathbb{Z}\} \right)$$

then for any given $\mathbf{r} \in \mathbb{R}^k$, we take $\mathbf{w} = \lfloor \mathbf{r} \rfloor + \sum_{i=1}^k \mathbb{1}(\mathbf{r}_i \geq \mathbf{f}_i + \lfloor \mathbf{r}_i \rfloor) \mathbf{e}^i \in \mathbb{Z}^k$, which is an integer vector such that $\bar{\mathbf{r}} := \mathbf{r} - \mathbf{w} \in [\mathbf{f}_1 - 1, \mathbf{f}_1] \times \cdots \times [\mathbf{f}_k - 1, \mathbf{f}_k]$. Therefore,

$$\pi_{\mathbf{f},\boldsymbol{\mu}}(\mathbf{r}) = \min_{\mathbf{z} \in \mathbb{Z}^n} \max_{j=0,\ldots,k} \langle \mathbf{a}^j, \mathbf{r} + \mathbf{z} \rangle$$

$$= \min_{\mathbf{z} \in \mathbb{Z}^n} \max_{j=0,\ldots,k} \langle \mathbf{a}^j, \bar{\mathbf{r}} + \mathbf{z} \rangle$$

$$= \min_{i \in \{1,\ldots,k\}} \min_{\lambda \in \mathbb{Z}} \max_{j=0,\ldots,k} \langle \mathbf{a}^j, \bar{\mathbf{r}} + \lambda \mathbf{e}^i \rangle$$

$$= \min_{i \in \{1,\ldots,k\}} \min_{\lambda \in \mathbb{Z}_+} \max_{j=0,\ldots,k} \langle \mathbf{a}^j, \bar{\mathbf{r}} + \lambda \mathbf{e}^i \rangle.$$

Notice that for any $i \in \{1, \ldots, k\}$,

$$\max_{j=0,\ldots,k} \langle \mathbf{a}^j, \bar{\mathbf{r}} + \lambda \mathbf{e}^i \rangle$$

$$= \max \left\{ \langle \mathbf{a}^0, \bar{\mathbf{r}} + \lambda \mathbf{e}^i \rangle, \frac{\bar{\mathbf{r}}_1}{\mathbf{f}_1 - 1}, \ldots, \frac{\bar{\mathbf{r}}_{i-1}}{\mathbf{f}_{i-1} - 1}, \frac{\bar{\mathbf{r}}_i + \lambda}{\mathbf{f}_i - 1}, \frac{\bar{\mathbf{r}}_{i+1}}{\mathbf{f}_{i+1} - 1}, \ldots, \frac{\bar{\mathbf{r}}_k}{\mathbf{f}_k - 1} \right\}$$

$$= \max \left\{ \frac{\sum_{j=1}^k \boldsymbol{\mu}_j \bar{\mathbf{r}}_j}{\sum_{j=1}^k \boldsymbol{\mu}_j \mathbf{f}_j} + \frac{\boldsymbol{\mu}_i}{\sum_{j=1}^k \boldsymbol{\mu}_j \mathbf{f}_j} \lambda, \frac{\bar{\mathbf{r}}_i}{\mathbf{f}_i - 1} + \frac{1}{\mathbf{f}_i - 1} \lambda, \max \left\{ \frac{\bar{\mathbf{r}}_j}{\mathbf{f}_j - 1} : j \in [k] \backslash \{i\} \right\} \right\} \quad (7)$$

is a one-dimensional piecewise linear function in $\lambda$ with at most three pieces (relaxing the feasible region of $\lambda$ from $\mathbb{Z}$ to $\mathbb{R}$). Using the notation in Algorithm 1, let $p = \sum_{j=1}^k \boldsymbol{\mu}_j \bar{\mathbf{r}}_j$, $q = \sum_{j=1}^k \boldsymbol{\mu}_j \mathbf{f}_j \geq \frac{1}{\tau} \sum_{j=1}^k \mathbf{f}_j > 0$ (since $\mathbf{f} \neq \mathbf{0}$), $a = \max \left\{ \frac{\bar{\mathbf{r}}_j}{\mathbf{f}_j - 1} : j \in \{1, \ldots, k\} \right\}$, $i^* = \arg\max \left\{ \frac{\bar{\mathbf{r}}_j}{\mathbf{f}_j - 1} : j \in \{1, \ldots, k\} \right\}$, $b = \max \left\{ \frac{\bar{\mathbf{r}}_j}{\mathbf{f}_j - 1} : j \in \{1, \ldots, k\} \backslash \{i^*\} \right\}$. Then for $i \in \{1, \ldots, k\}$, there are two cases:

**Case 1:** $i \neq i^*$. In this case, notice that for $\lambda \geq 0$ we have

$$a = \frac{\bar{\mathbf{r}}_{i^*}}{\mathbf{f}_{i^*} - 1} \geq \frac{\bar{\mathbf{r}}_i}{\mathbf{f}_i - 1} \geq \frac{\bar{\mathbf{r}}_i + \lambda}{\mathbf{f}_i - 1}$$

since $\mathbf{f}_i - 1 < 0$. Then,

$$\max_{j=0,\ldots,k} \langle \mathbf{a}^j, \bar{\mathbf{r}} + \lambda \mathbf{e}^i \rangle = \max \left\{ \frac{p + \boldsymbol{\mu}_i \lambda}{q}, \frac{\bar{\mathbf{r}}_i + \lambda}{\mathbf{f}_i - 1}, a \right\} = \max \left\{ \frac{p + \boldsymbol{\mu}_i \lambda}{q}, a \right\}$$

is a nondecreasing function of $\lambda$ over $[0, +\infty)$. Therefore,

$$\min_{\lambda \in \mathbb{Z}_+} \max \left\{ \frac{p + \boldsymbol{\mu}_i \lambda}{q}, a \right\} = \min_{\lambda \in \mathbb{R}_+} \max \left\{ \frac{p + \boldsymbol{\mu}_i \lambda}{q}, a \right\} = \max \left\{ \frac{p}{q}, a \right\}.$$

**Case 2:** $i = i^*$. In this case, we can rewrite (7) as

$$\max_{j=0,\ldots,k} \langle \mathbf{a}^j, \bar{\mathbf{r}} + \lambda \mathbf{e}^i \rangle = \max \left\{ \frac{p + \boldsymbol{\mu}_{i^*} \lambda}{q}, \frac{\bar{\mathbf{r}}_{i^*} + \lambda}{\mathbf{f}_{i^*} - 1}, b \right\}.$$

The slopes of these three affine linear functions are $\frac{\boldsymbol{\mu}_{i^*}}{q} \geq \frac{1}{\tau q} > 0$, $\frac{1}{\mathbf{f}_{i^*} - 1} < 0$, and $0$ respectively. The intersection point of the graphs of the first two functions is given by $J = \left( \lambda^*, \frac{\bar{\mathbf{r}}_{i^*} + \lambda^*}{\mathbf{f}_{i^*} - 1} \right)$, where $\lambda^* = \frac{q \bar{\mathbf{r}}_{i^*} - p(\mathbf{f}_{i^*} - 1)}{(\mathbf{f}_{i^*} - 1)\boldsymbol{\mu}_{i^*} - q}$ since

$$\frac{p + \boldsymbol{\mu}_{i^*} \lambda^*}{q} = \frac{\bar{\mathbf{r}}_{i^*} + \lambda^*}{\mathbf{f}_{i^*} - 1} \iff \underbrace{((\mathbf{f}_{i^*} - 1)\boldsymbol{\mu}_{i^*} - q)}_{< 0} \lambda^* = q\bar{\mathbf{r}}_{i^*} - p(\mathbf{f}_{i^*} - 1) \iff \lambda^* = \frac{q\bar{\mathbf{r}}_{i^*} - p(\mathbf{f}_{i^*} - 1)}{(\mathbf{f}_{i^*} - 1)\boldsymbol{\mu}_{i^*} - q}.$$

Observe that $\lambda^*$ is always an optimal solution to $\min_{\lambda \in \mathbb{R}} \max_{j=0,\ldots,k} \langle \mathbf{a}^j, \bar{\mathbf{r}} + \lambda \mathbf{e}^{i^*} \rangle$, regardless of whether the constant function $b$ is below or above $J$. Since the maximum of three affine linear functions forms a convex function, the optimal solution of $\min_{\lambda \in \mathbb{Z}} \max_{j=0,\ldots,k} \langle \mathbf{a}^j, \bar{\mathbf{r}} + \lambda \mathbf{e}^{i^*} \rangle$ is attained at either $\lfloor \lambda^* \rfloor$ or $\lceil \lambda^* \rceil$. Therefore, we have

$$\min_{\lambda \in \mathbb{Z}} \max_{j=0,\ldots,k} \langle \mathbf{a}^j, \bar{\mathbf{r}} + \lambda \mathbf{e}^{i^*} \rangle$$

$$= \min \left\{ \max_{j=0,\ldots,k} \left\langle \mathbf{a}^j, \bar{\mathbf{r}} + \lfloor \lambda^* \rfloor \mathbf{e}^{i^*} \right\rangle, \max_{j=0,\ldots,k} \left\langle \mathbf{a}^j, \bar{\mathbf{r}} + \lceil \lambda^* \rceil \mathbf{e}^{i^*} \right\rangle \right\}$$

$$= \min \left\{ \max \left\{ \frac{p + \boldsymbol{\mu}_{i^*} \lceil \lambda^* \rceil}{q}, \frac{\bar{\mathbf{r}}_{i^*} + \lceil \lambda^* \rceil}{\mathbf{f}_{i^*} - 1}, b \right\}, \max \left\{ \frac{p + \boldsymbol{\mu}_{i^*} \lfloor \lambda^* \rfloor}{q}, \frac{\bar{\mathbf{r}}_{i^*} + \lfloor \lambda^* \rfloor}{\mathbf{f}_{i^*} - 1}, b \right\} \right\}.$$

Thus, take the minimum of the two cases, we have

$$\pi_{\mathbf{f}, \boldsymbol{\mu}}(\mathbf{r}) = \min \left\{ \max \left\{ \frac{p + \boldsymbol{\mu}_{i^*} \lceil \lambda^* \rceil}{q}, \frac{\bar{\mathbf{r}}_{i^*} + \lceil \lambda^* \rceil}{\mathbf{f}_{i^*} - 1}, b \right\}, \max \left\{ \frac{p + \boldsymbol{\mu}_{i^*} \lfloor \lambda^* \rfloor}{q}, \frac{\bar{\mathbf{r}}_{i^*} + \lfloor \lambda^* \rfloor}{\mathbf{f}_{i^*} - 1}, b \right\}, \max \left\{ \frac{p}{q}, a \right\} \right\}.$$

$\square$

*Proof of Proposition 4.3.* Observe that the lifting region

$$R \subseteq G(\mathbf{f}, \boldsymbol{\mu}) \subseteq \bigcup_{i=1}^{k} \left( [\mathbf{f}_1 - 1, \mathbf{f}_1] \times \cdots \times [\mathbf{f}_k - 1, \mathbf{f}_k] + \left\{ \lambda \mathbf{e}^i : \lambda \in \left[ 0, \frac{1}{\boldsymbol{\mu}_i} \right] \right\} \right)$$

$$\subseteq \bigcup_{i=1}^{k} \left( [\mathbf{f}_1 - 1, \mathbf{f}_1] \times \cdots \times [\mathbf{f}_k - 1, \mathbf{f}_k] + \left\{ \lambda \mathbf{e}^i : \lambda \in [0, \tau] \right\} \right)$$

$$= \bigcup_{i=1}^{k} \left( [\mathbf{f}_1 - 1, \mathbf{f}_1] \times \cdots \times [\mathbf{f}_k - 1, \mathbf{f}_k] + \left\{ \lambda \mathbf{e}^i : \lambda \in \{0, 1, \ldots, \tau\} \right\} \right).$$

Let $\bar{\mathbf{r}}^j = \mathbf{r}^j - \lfloor \mathbf{r}^j \rfloor - \sum_{i=1}^{k} \mathbb{1}(\mathbf{r}_i^j \geq \mathbf{f}_i + \lfloor \mathbf{r}_i^j \rfloor) \mathbf{e}^i \in [\mathbf{f}_1 - 1, \mathbf{f}_1] \times \cdots \times [\mathbf{f}_k - 1, \mathbf{f}_k]$, $j = \{1, \ldots, n\}$. Then for any fixed $j$, based on the proof of Theorem 4.2, we have

$$\pi_{\mathbf{f}, \boldsymbol{\mu}}(\mathbf{r}^j) = \min_{i \in \{1, \ldots, k\}} \min_{\lambda \in \{0, 1, \ldots, \tau\}} \max_{l=0, \ldots, k} \langle \mathbf{a}^l, \bar{\mathbf{r}}^j + \lambda \mathbf{e}^i \rangle$$

$$= \min_{i \in \{1, \ldots, k\}} \min_{\lambda \in \{0, 1, \ldots, \tau\}} \max \left\{ \frac{p_j + \boldsymbol{\mu}_i \lambda}{q}, \frac{\bar{\mathbf{r}}_i^j + \lambda}{\mathbf{f}_i - 1}, \max \left\{ \frac{\bar{\mathbf{r}}_l^j}{\mathbf{f}_l - 1} : l \in \{1, \ldots, k\} \setminus \{i\} \right\} \right\}$$

$$= \min \left\{ \max \left\{ \frac{p_j}{q}, a_j \right\}, \min_{\lambda \in \{0, 1, \ldots, \tau\}} \max \left\{ \frac{p_j + \boldsymbol{\mu}_{i_j^*} \lambda}{q}, \frac{\bar{\mathbf{r}}_{i_j^*}^j + \lambda}{\mathbf{f}_{i_j^*} - 1}, b_j \right\} \right\},$$

where, as defined in Algorithm 1 and Theorem 4.2, $p_j = \sum_{i=1}^{k} \boldsymbol{\mu}_i \bar{\mathbf{r}}_i^j$, $q = \sum_{i=1}^{k} \boldsymbol{\mu}_i \mathbf{f}_i \geq \frac{1}{\tau} \sum_{i=1}^{k} \mathbf{f}_i > 0$, $a_j = \max \left\{ \frac{\bar{\mathbf{r}}_i^j}{\mathbf{f}_i - 1} : i \in \{1, \ldots, k\} \right\}$, $i_j^* = \arg\max \left\{ \frac{\bar{\mathbf{r}}_i^j}{\mathbf{f}_i - 1} : i \in \{1, \ldots, k\} \right\}$, $b_j = \max \left\{ \frac{\bar{\mathbf{r}}_i^j}{\mathbf{f}_i - 1} : i \in \{1, \ldots, k\} \setminus \{i_j^*\} \right\}$.

This motivates considering the following numbers, for any fixed $j \in \{1, \ldots, n\}$:

$$\frac{p_j}{q}, a_j, \frac{p_j + \boldsymbol{\mu}_{i_j^*} \lambda}{q}, \frac{\bar{\mathbf{r}}_{i_j^*}^j + \lambda}{\mathbf{f}_{i_j^*} - 1}, b_j, \lambda \in \{0, 1, \ldots, \tau\}.$$

There are $1 + 1 + (\tau + 1) + (\tau + 1) + 1 = 2\tau + 5$ numbers in total, so the pairwise comparison of these numbers can be done by at most $(2\tau + 5)(2\tau + 6)/2 \leq 2(\tau + 3)^2$ hyperplanes in the $\boldsymbol{\mu}$ space $\Delta_k^\tau$. This is because, in the worst scenario, the equality

$$\frac{p_j + \boldsymbol{\mu}_{i_j^*} \lambda}{q} = \frac{\bar{\mathbf{r}}_{i_j^*}^j + \lambda'}{\mathbf{f}_{i_j^*} - 1} \iff p_j + \lambda \boldsymbol{\mu}_{i_j^*} - \frac{\bar{\mathbf{r}}_{i_j^*}^j + \lambda'}{\mathbf{f}_{i_j^*} - 1} q = 0$$

is a hyperplane on $\boldsymbol{\mu}$, and similar arguments apply to other pairs. These hyperplanes decompose the $\Delta_k^\tau$ space into some regions such that, within each region, the order of the above $2\tau + 5$ numbers is fixed. Within each region, $\pi_{\mathbf{f}, \boldsymbol{\mu}}(\mathbf{r}^j)$ can be expressed as the quotient of two affine linear functions in $\boldsymbol{\mu}$, with the denominator being $q = \sum_{i=1}^{k} \boldsymbol{\mu}_i \mathbf{f}_i > 0$. Overlapping all hyperplanes across $j \in \{1, \ldots, n\}$ results in at most $2n(\tau + 3)^2$ hyperplanes decomposing the $\Delta_k^\tau$ space.

□

*Proof of Theorem 4.4.* Similar to what we have done in the proof of Theorem 3.2, let $\xi_1, \ldots, \xi_\Gamma$ denote the degree 5 polynomials given in Lemma A.2, where $\Gamma = \mathcal{O}\left( (14)^n (m + 2n)^{3n^2} \varrho^{5n^2} \right)$. By Proposition 4.3, there are at most $2n(\tau + 3)^2$ hyperplanes decomposing the $\Delta_k^\tau$ space such that, within each decomposed region, each coordinate of the cutting plane derived from the cut generating functions $\pi_{\mathbf{f}, \boldsymbol{\mu}}$ is a fixed rational function given by the quotient of two affine linear functions on $\boldsymbol{\mu}$. Also, by Lemma A.1, these hyperplanes decompose the $\boldsymbol{\mu}$ space $\Delta_k^\tau$ into at most

$$\left( \frac{2n(\tau + 3)^2}{k} \right)^k \leq 2^k n^k (\tau + 3)^{2k} := K$$

regions, denoted as $Q_1, \ldots, Q_{\widetilde{K}}$, where $\widetilde{K} \leq K$.

We fixed a region $Q_i$ for any $i \in \{1, \ldots, \widetilde{K}\}$. For any $\boldsymbol{\mu} \in Q_i$, we claim that $\begin{bmatrix} \boldsymbol{\alpha}(\boldsymbol{\mu}) \\ \beta(\boldsymbol{\mu}) \end{bmatrix}$ can always be expressed as $\frac{\mathbf{p}^i(\boldsymbol{\mu})}{q(\boldsymbol{\mu})} \in \mathbb{R}^{n+1}$, where each coordinate of $\mathbf{p}^i$ is an affine linear function of $\boldsymbol{\mu}$, and $q$ is a fixed affine linear function of $\boldsymbol{\mu}$. This is because, according to the form derived in Proposition 4.3, the values of the trivial liftings are either rational functions with the denominator $q(\boldsymbol{\mu}) = \sum \boldsymbol{\mu}_i \mathbf{f}_i > 0$ (which is fixed once the instance $I = (A, \mathbf{b}, \mathbf{c})$ is fixed), or just some constants independent of $\boldsymbol{\mu}$. Then by Lemma A.3, with a fixed $I = (A, \mathbf{b}, \mathbf{c})$, the cutting plane for the original problem is a fixed affine transformation of the above form. Therefore, in this fixed region $Q_i$, the final form of the cutting plane is fixed and can be written as $\frac{\mathbf{p}^i}{q}(\boldsymbol{\mu}) \in \mathbb{R}^{n+1}$. Therefore,

$$\left( \xi_1 \left( \begin{bmatrix} \boldsymbol{\alpha}(\boldsymbol{\mu}) \\ \beta(\boldsymbol{\mu}) \end{bmatrix} \right), \ldots, \xi_\Gamma \left( \begin{bmatrix} \boldsymbol{\alpha}(\boldsymbol{\mu}) \\ \beta(\boldsymbol{\mu}) \end{bmatrix} \right) \right) = \left( \left( \xi_1 \circ \frac{\mathbf{p}^i}{q} \right)(\boldsymbol{\mu}), \ldots, \left( \xi_\Gamma \circ \frac{\mathbf{p}^i}{q} \right)(\boldsymbol{\mu}) \right)$$

gives $\Gamma$ fixed rational functions of $\boldsymbol{\mu}$, decomposing $Q_i$, in the form of the quotient of two degree 5 polynomials. For any $\boldsymbol{\mu} \in Q_i$ within each decomposed region, these $\Gamma$ rational functions have an invariant sign pattern, hence the tree size after adding $\boldsymbol{\alpha}(\boldsymbol{\mu})^\mathsf{T} \mathbf{x} \leq \beta(\boldsymbol{\mu})$ at the root remains the same. By traversing all $Q_i$, the total number of such rational functions is given by

$$\mathcal{O} \left( 2^k n^k (\tau + 3)^{2k} (14)^n (m + 2n)^{3n^2} \varrho^{5n^2} \right),$$

then the pseudo-dimension result follows from Lemma 2.1:

$$\mathrm{Pdim} \left( \{ T^k(\cdot, \boldsymbol{\mu}) : \mathcal{I} \to [0, B] \mid \boldsymbol{\mu} \in \Delta_k^\tau \} \right) = \mathcal{O} \left( k n^2 \log((m+n)\varrho) + k^2 \log(n\tau) \right).$$

$\square$

