# OpenReview forum: "Learning Cut Generating Functions for Integer Programming"
_NeurIPS.cc/2024/Conference — NeurIPS 2024 poster_

### Official Review · Reviewer_6RAE · 2024-07-12

**Soundness:** 2
**Presentation:** 3
**Contribution:** 2
**Rating:** 5
**Confidence:** 2

**Summary:**

The paper is concerned with the interplay of learning theory and the branch-and-cut algorithm for solving mixed-integer programs (MIPs).
Concretely the authors analyse the problem of cut selection.
In general, cut selection asks for a given (class of) MIPs: What cut(s) should be added to the lp relaxation, to speed up the branch-and-bound algorithm as much as possible? The authors limit the scope of this board question to:
How should one parameterize a concrete cut-generating function, to generate a single cut, which, when added to the problem at the root node, minimizes the number of branch-and-bound nodes the solver has to create?
For the selection of a cut generating function, the authors list three criteria:
1: There should be an efficient way, to compute the cutting planes 2: One should be able to prove concrete sample complexity bounds 3: One should be able to demonstrate that the cutting planes are “significantly better in practice than classical cuts”
The authors then proof sample complexity bounds for three different settings.
1) One wants to find the parameters for the Gomory and Johnson [2003] cut generating function (which generates cuts from a single row of the simplex tableau) minimizing the expected tree size for instances drawn from $D$.
2) One wants to find the parameters for a subfamily of cut generating functions studied by Basu and Sankaranarayan [2019]  (which generate cuts from k many rows of the simplex tableau) again minimizing the expected tree size for instances drawn from $D$.
3) One wants to train a neural network to choose a good instance (not distribution) dependent cut generating function.

In their numerical experiments, the authors investigate the effect of the cut generating functions on some generated knapsack and partition problems. The goal is to find evidence for better performance of the chosen cut generating functions when compared to classical (here GMI only) cuts. For 1-d knapsack, the cut generating function is shown to clearly outperform GMI cuts in the chosen setting. For all other instances, GMI performs worse than at least one cut generating function, but the difference is smaller, especially for the packing instances.

**Strengths:**

The paper is well-structured and easy to read.
The provided code allows for the reproducibility of nearly all experiments (only for knapsack_50 I get “index 0 is out of bounds for axis 0 with size 0” and no results, even if I run the code to regenerate the instances).

**Weaknesses:**

The main weakness I see is the little empirical evidence for some strong claims about the  Gomory and Johnson [2003] cut generating functions. See e.g. line 220 and 251: “and they result in significantly smaller tree sizes compared to traditional cutting planes”. This statement seems over the top.
1) the numerical experiments do not always indicate “significant”
2) the only tested classical baseline are GMI cuts, which are by far not the only class of cuts doing heavy lifting in modern branch-and-cut solvers
3) for the instances on which the impact is high (1-d Knapsack), I think the combination of setting and instance class, does not allow general claims, see below.

L 316: “Instances were generated using the distribution proposed by Chvátal in Chvátal [1980], also used in Balcan et al. [2021b].”
-> this is not correct. The authors use the same parameters as Balcan et al., but they are at best related to Chvátal instances. Chvátal explicitly designs instances, which are provably hard to solve for certain branch-and-bound algorithms.
To achieve this, all constraint coefficients $a_{ij}$ are sampled from $[1, 10^{(n/2)}]$ and $b_j = floor(\sum (a_i / 2))$.
Balcan et al. sample a_ij from a uniform normal distribution $N(50, 2)$. This leads to extremely similar a_ij (so quite far from Chvátals distribution) values and makes the instances extremely easy in practice. For 1-dimensional knapsack the original constraint of the problem formulation is tight. When turning on primal heuristics (which are turned off by the authors, but not by Balcan et al.) Gurobi solves all instances at the root node, before even solving the relaxation. From my perspective, there is little to learn from experiments on these types of instances (1-d knapsack).
I am well aware, that the authors focus on theoretical aspects of machine learning and cut selections, but when explicitly listing “3) we should be able to demonstrate that these new cutting planes are significantly better in practice than classical cuts” (line 109) as one of three requirements for the cut generating functions, I expect stronger empirical evidence.

**Questions:**

Additional and minor remarks:
- L 6: “optimal cutting planes”. “Optimal” is very global and strict. Maybe “good”
- L 24: “small representative sample”. Representative seems subjective
- L 36: “we understand many of their theoretical properties”. “We” sounds ambiguous here. Maybe “many theoretical properties are understood”
- L 53: “with several excellent insights”. “Excellent” is quite subjective
- L 144: Z^n should be Z^k
- Figure 1: The use of the label “y” for the y-Axis is confusing to me
- Figure 1: For r = 1 the graph has a red circle, indicating it is not defined here. The same should hold for r=0?
- Figure 1: The font seems unnecessarily small
- L 296: If “their” in “A direct applications of their main theorem” refers to Cheng et al. [2024] from line 278, I feel like the reference is too far away, for the reader to parse “their” in the context of Cheng et al.
- L 314: The two types of instances under consideration are quite similar (in referenced Tang 2020: “A knapsack problem is a binary packing problem”.) I think it might make sense to mention their similarity and the fact, that one is integer and one is binary
- Table 1: For $d$-dimensional knapsack, there are $d$ constraints significantly different from the remaining $n$. Are these constraints considered first, for k-row? Does this explain why e.g. 2-row is best for 2-d knapsack?
- Small and capital letter inconsistency in references. E.g. 352 & 358: “cambridge university press” and “Cambridge University Press”

**Limitations:**

The author's discussion of their limitations is fine, apart from the part about the numerical experiments (see weaknesses).

---

> ### Author Rebuttal · Authors · 2024-08-06
>
> We greatly appreciate your thorough review and the thoughtful feedback. Regarding the experimental evidence, we acknowledge that our paper focuses primarily on theoretical aspects, and thus, the experimental section is somewhat preliminary. First, we agree that it was poor scholarship on our part to call our distribution the Chvátal distribution. Our distribution is indeed the same as what was done in the Balcan paper. We will reword to something like "what Balcan et al call the Chvátal distribution".
>
> However, we believe that even these basic experiments lend support to our claim that some of the CGFs we consider can result in significantly smaller tree sizes compared to some traditional cutting planes. Our reasoning is as follows:
>
> 1. Even though the 1D knapsack problem is very simple, this does not detract from the theoretical interest of the result. Specifically, when using only branch-and-bound for certain specific distributions (such as the "Chvátal distribution" considered here), there are indeed some CGFs that perform significantly better than GMI cuts.
>
> 2. Even if we disregard the 1D knapsack results, we believe the other experiments on multidimensional knapsacks and packing problems also give some support to our claim:
>     - Regardless of the problem type or the number of rows used to generate cutting planes, we selected a parameter from only 121 random candidate CGF parameters, and for one-row cuts, we only considered the simple case of $p = q = 2$, resulting in only two parameters. Despite this, some CGFs performed better, which demonstrates the potential of CGFs.
>     - In the last column, we provided the "best 1-row cut" for the one-dimensional CGF, obtained by enumerating and selecting the best 1D CGF for each instance. We believe this result shows significant improvement over GMI cuts, even for the packing problem settings. While this method is not really "learning," we believe it does demonstrate the potential of CGFs. Also, for this method, in Section 5 we proved the learnability and rigorous sample complexity for solving the problem of "selecting good CGFs for each instance" using neural networks, which may provide some insights to practitioners to develop some neural network based instance-specific selection methods.
>
> 3. We compare with GMI cuts because this is considered a "gold standard" in evaluating cutting planes in IP computational literature. While there are indeed several other cutting plane families that are used in modern solvers, the first test for any new family of cutting planes (from a practical perspective) is usually to see how well they do in comparison to GMIs. They are arguably the most important and popular *general purpose* cutting planes, adopted by commercial solvers for almost thirty years now (Conforti et al., 2014). Additionally, GMI cutting planes are generated by an extreme CGF (Section 2.1 has the corresponding CGF definition). Given that the main focus of the paper is on the theory developed, we wanted to keep the message of our experimental section crisp and streamlined. Comparisons with GMI seemed like a good way to achieve this.
>
> We do agree that perhaps our choice of wording in the experimental section (and in references to it) is overenthusiastic. We will reword and tone down some of the language. The reason behind our excitement is that we have not seen such promising results with CGFs in the computational IP literature. We concede that some of instances are not the most challenging ones for modern solvers. Nevertheless, we do not know of any published (or folklore) cases of instances where CGFs have previously been demonstrated to have such stark improvements over GMI cuts. And this is even more true for the Gomory-Johnson family of CGFs. While there are some computational results showing some promise with CGFs coming from lattice-free sets and their liftings, to the best of our knowledge, the Gomory-Johnson families have not been shown to do better than GMIs (beyond some artifical instances) in any previous study. Observing an improvement with these CGFs that was quite a contrast from previous computational experience was the basis for our enthusiastic wording. One reason could indeed be the choice of instances. Nevertheless, we think another reason is that previous work focused more on the gap closed at the root, as opposed to the overall tree size. We hope that our preliminary results convinces the community to revisit these CGFs, especially with the view to investigate their effect on the overall tree sizes (both theoretically and practically). We will also expand our computational investigation to look at other families of instances to see if the improvement in tree size is as dramatic as we observed on these knapsack instances.
>
> To summarize, our hope with the experimental section is to "whet the IP community's appetite" and rejuvenate its interest in CGFs. We believe they can be useful, and using learning techniques may be a good way to unlock their true potential.
>
> Some responses to your more detailed questions:
>
> - > Z^n should be Z^k
>
>     It should be $\mathbb{Z}^n$ since $\mathbf{r}^1,...,\mathbf{r}^n \in \mathbb{R}^k$ are vectors.
>
> - > Figure 1: For $r=1$ ... should hold for $r=0$?
>
>     The plots in Figure 1 are on the interval $[0,1)$, so $r = 1$ is not included, which is necessary because the generating function of Chvátal-Gomory cut, $\text{CG}_f(r)$, is not continuous at $r \in \mathbb{Z}$. However, $r = 0$ is defined, and by periodicity, for these functions, the value at $r = 1$ is the same as the value at $r = 0$.
>
> - > Table 1: For $d$-dimensional ..., 2-row is best for 2-d knapsack?
>
>     Yes, this is a very insightful point. We share the same intuition. Indeed, for $k$-row cuts, we selected the first $k$ rows from the simplex tableau to generate a cut, so it makes sense that the 2-row cut works best for the 2D knapsack problem.
>
> We will address the other minor remarks in the final version of the paper.

---

### Official Review · Reviewer_TAJ6 · 2024-07-12

**Soundness:** 3
**Presentation:** 4
**Contribution:** 3
**Rating:** 7
**Confidence:** 2

**Summary:**

This paper studies the learning of generic classes of cut generating functions, which can be used as an algorithmic tool for solving integer programming problems. The paper presents a handful of cut generating function families, studies the learning complexity of these functions, and presents a computational study of those CGF families on standard integer programming problems.

**Strengths:**

The setting is of clear interest to the integer programming (IP) community. I cannot vouch for the novelty or correctness of the learning-theoretic content of the paper, but the IP content is crisp and clean, and the high-level idea and computational setup all make sense to me.

**Weaknesses:**

I feel like there is some connective tissue missing from the paper, particularly joining the computational study with the rest of the paper. Section 5 seems like an interesting connection to draw, but does not appear in the computational study and feels a bit disjointed from the rest of the paper.

And while the paper spends much of its effort to show that the cut generating functions _can_ be learned, the computational study is relatively rudimentary and does not really attempt to develop very sophisticated techniques to actually learn the CGFs in practice (to be clear, this is a reasonable tradeoff to make for a theoretical paper).

**Questions:**

None.

**Limitations:**

Yes.

---

> ### Author Rebuttal · Authors · 2024-08-06
>
> Thank you very much for the thoughtful and encouraging review. Section 5 is meant to illustrate that our analysis extends to the setting where one wishes to select cutting planes tailored to instances using, say, a neural network mapping from instances to cutting planes. This was done in a more general algorithm design setting by Cheng et al (2024), but the analysis can be easily adapted to CGFs. This can potentially lead to stronger gains since the CGFs are tailored to specific isntances, as opposed to using the same CGF for all instances.

---

### Official Review · Reviewer_yCxh · 2024-07-14

**Soundness:** 3
**Presentation:** 3
**Contribution:** 3
**Rating:** 7
**Confidence:** 3

**Summary:**

This work presents sample complexity results for learning parameters for certain classes of cut generating functions, along with some numerical experiments. These are functions that determine coefficients of cutting planes to help solve mixed-integer programming problems, and some of the most effective cutting planes in practice (e.g. GMI cuts) can be expressed as cut generating functions. More specifically, this paper proves pseudo-dimension bounds on an established class of one-dimensional cut generating functions, and on a class of k-dimensional functions, both of which generalize GMI cuts. This latter function can be (non-trivially) computed efficiently. Furthermore, the authors provide pseudo-dimension bounds for these functions for the case where the parameters are learned by a neural network. Finally, the paper provides numerical experiments to highlight the learnability of parameters for these families of cuts on small instances.

**Strengths:**

This work adds an interesting and novel learning-theoretical perspective to the cut generating function literature. Making general cut generating functions practical has always been challenging, and this paper suggests potential in learning them in a principled and theoretically grounded fashion. While we are still far from being able to use cut generating functions in the way that the paper promotes, this is a valuable step that advances the field further and I believe that this can inform future learning-based cut generation methods. The paper is overall written in a clear way. I checked the proofs at a high level but could not do so in detail since my familiarity with learning theory is limited. The computational results, while focused on small cases, show good potential for learning-based methods.

**Weaknesses:**

None of the weaknesses that I see are particularly major. I would have liked to see further experiments on more realistic scenarios, but I understand that we can derive insights from a very focused experimental setup and the theoretical contributions are the main focus of this paper. I leave specific issues for the Questions section below.

**Questions:**

These are all minor comments.

1. It would be nice to add the geometrical interpretation of the parameters of the functions to provide a quicker understanding for the reader. It takes a little while by looking at the functions and the examples (which are helpful) to understand, and for me it was easier to understand some of the proofs after I understood what those parameters were doing. You can for example add them with Figure 2 in the Appendix.

2. I am not sure I can visualize the k-dimensional function very well. Would you be able to explain what motivates this particular function, and perhaps include 3d examples for the case where k = 2 (like you did with k = 1)?

3. Is the definition of tree size here the worst-case tree size across variable and node selection? If relevant, could you add a precise definition of tree size to the paper?

4. Could you add to the paper why these cut generating functions are extreme? I believe those come from previous work (e.g. the 1-row one has two slopes)?

5. I am confused as to how you have 5-row cuts and 10-row cuts for knapsack with 2 and 3 rows. If I missed something, please add the explanation to the paper.

6. This question is mainly out of curiosity (though could be interesting to discuss in the paper if you have a good answer): I see that increasing the number of rows sometimes help, but generally do not seem to be too helpful. It may be more difficult to find a good multi-row cut with more rows but a fixed number of samples. Do you have any sense on how the quality of your cuts would improve with more samples?

7. Please fix the citation types (those missing parenthesis).

**Limitations:**

There is not much discussion on limitations, though the paper is mostly theoretical. Perhaps the paper could discuss some more what these results could lead to in future work, and limitations of interpreting these small-scale experiments.

---

> ### Author Rebuttal · Authors · 2024-08-06
>
> Thank you very much for the insightful review and providing detailed feedback. Regarding your questions:
>
> 1. This is an good point. The two families of cut-generating functions considered in this paper are indeed parametrized geometrically. We will include additional explanations and figures to clarify this further.
>
> 2. The $k$-dimensional cut-generating function in this paper is a subfamily of those studied by Basu and Sankaranarayanan (2019). Specifically, it is the trivial lifting of the gauge function of a family of maximal $(\mathbf{f} + \mathbb{Z}^k)$-free simplices, defined as $\mathbf{f} + \lbrace \mathbf{x} \in \mathbb{R}^k : \sum_{i=1}^{k} \mu_i \mathbf{x}_i \leq 0, \mathbf{x}_1 \geq -1, \dots, \mathbf{x}_k \geq -1 \rbrace$, parametrized by $\mu$ restricted to the standard $k$-dimensional simplex. We indeed have some 3D-printed models of these generating functions, as shown in our global response (see the attached PDF). We will include these figures in the paper.
>
> 3. Our main results are based on the piecewise structural results of the branch-and-cut tree size studied by Balcan et al. (2022), so we adhere to the same methods and assumptions for building branch-and-cut trees as in their paper. Therefore, the results apply to product scoring policy or lexicographic policy for variable selection and depth-first search policy for node selection. Using the same technique, one can prove similar results for other selection policies, such as the best-bound policy for node selection. We will clarify this further in the paper.
>
> 4. Gomory and Johnson (2003) proved that the one-dimensional cut-generating functions (CGFs) considered in this paper are extreme. For $k$-dimensional CGFs, they are minimal valid functions as they are the trivial lifting of the gauge function of maximal $(\mathbf{f} + \mathbb{Z}^k)$-free convex sets with the covering property (Basu et al., 2013; Conforti et al., 2014; Averkov and Basu, 2015). Then, these $k$-dimensional CGFs are extreme by the $(k+1)$-slope theorem (Basu et al., 2011). We will include these explanations in the appendix of the paper.
>
> 5. There are some trivial constraints for those $0/1$ knapsack problems restricting the decision variables to be no larger than 1. We do not explicitly handle these upper bounds differently, and consequently, they contribute to the simplex tableaux.
>
> 6. This is a very interesting and important question. There are two related issues here. One is the classical bias-variance/bias-complexity/overfitting-underfitting tradeoff in any statistical method such as ours. The second issue is the difficulty in finding the optimal solution to the ERM problem $\min_{{\mu}} \frac{1}{N}\sum_{i=1}^N T^k(I_i, {\mu})$, where $T^k(I_i,{\mu})$ is the branch-and-cut tree size of the instance $I_i$ after adding a cutting plane induced by a $k$-dimensional CGF parameterized by ${\mu}$ (i.e., $k$ is the number of rows used to generate the cutting plane).
>
>     Increasing the number of rows reduces the bias of the model, i.e., we expect the overall expected error to go down with the optimal choice of parameters ${\mu}$ (for minimizing the overall expected error). However, it increases the variance/complexity of the learning procedure. As you point out, if we fix the number of samples, increasing the number of rows leads to weaker guarantees on the expected error of the learned parameters, i.e., the ERM solution ${\hat\mu}$. Sample complexity tracks this trade-off quantitatively: it tells us how the sample size should grow if we increase the complexity of our model (i.e., increase the number of rows $k$ used to generate the cutting plane), if we wish to keep the same error and high probability guarantees.
>
>     From an empirical perspective of solving the ERM problem, the phenomenon you mentioned is the following. In the experiments of this paper, regardless of the value of $k$, we uniformly sampled a constant number of CGF parameters on the $k$-dimensional standard simplex and chose the best one. Therefore, for higher dimensions, the probability is lower that some of these randomly sampled points are good enough to solve the ERM problem. In this setting, increasing the number of sampled candidate parameters as $k$ increases is a reasonable way to improve the performance of the $k$-dimensional CGF. However, the number of candidate parameters might grow exponentially with $k$ for a provable guarantee. We have not theoretically or computationally explored this dependence.
>
>     Therefore, with more IP instance samples, we might consider moving away from this simple yet robust enumeration method and adopting heuristic algorithms to solve the ERM problem. For instance, as suggested by Cheng et al. (2024) in their paper that also involves the optimization of a similar ERM problem, we could use some reinforcement learning (RL) algorithms, treating the CGF parameters as continuous actions in the RL setting, to find a relatively good parameter setting.
>
> 7. Thank you for catching this. We will fix the citation types in the paper.

---

> > ### Comment · Reviewer_yCxh · 2024-08-10
> >
> > Thank you for the response. This answers all my questions and I appreciate the changes to the paper. I have read all reviews and rebuttals and I will keep my score. I am not too concerned with the limited computational experiments given the nature of this paper, and while I agree with the main concern of 6RAE, in my opinion it should be sufficient to change the language so that it is clearer that this is more of a scientific study rather than a practical one.

---

### Author Rebuttal · Authors · 2024-08-06

The attached PDF includes figures of a 2-dimensional cut generating function, related to reviewer yCxh's question (point 2).

---

### Decision · Program_Chairs · 2024-09-25

**Decision:**

Accept (poster)

**Comment:**

In this submission branch-and-cut algorithms are studied in a data-driven framework. For this, parameterized families of cut generating functions are considered and their sample complexity is analyzed. One assumes that problem instances are generated according to some unknown probability distribution and the question is how many samples have to be observed to find values for the parameters of the cut generating functions such that the expected size of the truncated branch-and-cut tree is approximately minimal. It is well-known that this question can be reduced to bounding the pseudo-dimension of the associated function class.

The main theoretical contribution is the analysis of the pseudo-dimension of different classes of cut generating functions, leading to upper bounds on the sample complexity. The submission concludes by an experimental evaluation for the multiple knapsack problem and packing problems. These experiments show that learning the parameters of the cut generating functions can indeed lead to be smaller branch-and-cut trees than the classical Gomory's mixed integer cut.

The reviewers think that this line of research is interesting and that the submission makes a strong theoretical contribution. The reviewers, however, also agree that it would be interesting to extend the experimental study, which seems somewhat preliminary at the moment.